RESEARCH COMMUNICATION

# Mitochondrial biogenesis is transcriptionally repressed in lysosomal lipid storage diseases

King Faisal Yambire[1,2,3], Lorena Fernandez-Mosquera[1], Robert Steinfeld[4], Christiane Mühle[5], Elina Ikonen[6], Ira Milosevic[3], Nuno Raimundo[1]*

[1]Institute of Cellular Biochemistry, University Medical Center Goettingen, Goettingen, Germany; [2]International Max-Planck Research School in Neuroscience, Goettingen, Germany; [3]European Neuroscience Institute Goettingen, University Medical Center Goettingen, Goettingen, Germany; [4]Klinik für Kinder- und Jugendmedizin, University Medical Center Goettingen, Goettingen, Germany; [5]Department of Psychiatry and Psychotherapy, Friedrich-Alexander University Erlangen-Nürnberg (FAU), Erlangen, Germany; [6]Department of Anatomy, Faculty of Medicine, University of Helsinki, Biomedicum Helsinki, Helsinki, Finland

*For correspondence: nuno.raimundo@med.uni-goettingen.de

**Competing interests:** The authors declare that no competing interests exist.

**Abstract** Perturbations in mitochondrial function and homeostasis are pervasive in lysosomal storage diseases, but the underlying mechanisms remain unknown. Here, we report a transcriptional program that represses mitochondrial biogenesis and function in lysosomal storage diseases Niemann-Pick type C (NPC) and acid sphingomyelinase deficiency (ASM), in patient cells and mouse tissues. This mechanism is mediated by the transcription factors KLF2 and ETV1, which are both induced in NPC and ASM patient cells. Mitochondrial biogenesis and function defects in these cells are rescued by the silencing of KLF2 or ETV1. Increased ETV1 expression is regulated by KLF2, while the increase of KLF2 protein levels in NPC and ASM stems from impaired signaling downstream sphingosine-1-phosphate receptor 1 (S1PR1), which normally represses KLF2. In patient cells, S1PR1 is barely detectable at the plasma membrane and thus unable to repress KLF2. This manuscript provides a mechanistic pathway for the prevalent mitochondrial defects in lysosomal storage diseases.

**Editorial note:** This article has been through an editorial process in which the authors decide how to respond to the issues raised during peer review. The Reviewing Editor's assessment is that all the issues have been addressed (see decision letter).

DOI: https://doi.org/10.7554/eLife.39598.001

## Introduction

Lysosomal storage diseases are a group of severe diseases caused by mutations in genes encoding for lysosomal proteins, and are referred to as storage diseases because one common phenotype is the accumulation of undigested substrates in the lysosomes, with the consequent enlargement and loss of function of the organelle (*Parenti et al., 2015*). The lysosomes have far-reaching roles beyond the 'recycling bin' paradigm, and are key players in nutrient sensing and metabolic regulation (*Ballabio, 2016*; *Lim and Zoncu, 2016*; *Settembre et al., 2013*). Furthermore, lysosomes are essential for the process of macroautophagy, and thus for the selective autophagy of mitochondria, the main mechanism to degrade dysfunctional mitochondria (*Pickles et al., 2018*). Mitochondrial perturbations have been widely reported in several lysosomal storage diseases (*Platt et al., 2012*; *Plotegher and Duchen, 2017*), including neuronal ceroid lipofuscinosis, Gaucher and Niemann-Pick diseases (*Jolly et al., 2002*; *Lim et al., 2015*; *Osellame et al., 2013*; *Torres et al., 2017*;

Woś et al., 2016). Nevertheless, it remains unclear why mitochondrial dysfunction is so prevalent in lysosomal storage diseases.

In this study, we focus on two lysosomal storage diseases, Niemann-Pick type C (NPC) and acid sphingomyelinase (ASM) deficiency. NPC is caused by mutations in the gene NPC1 or, less commonly, NPC2 (Patterson and Walkley, 2017; Schuchman and Wasserstein, 2016). NPC1 and NPC2 encode proteins involved in sphingomyelin and cholesterol efflux from the lysosome (Platt, 2014). ASM deficiency, also known as Niemann-Pick A/B, is caused by mutations in the gene SMPD1 encoding acid sphingomyelinase. ASM catalyzes the breakdown of sphingomyelin into ceramide and phosphorylcholine (Schuchman and Wasserstein, 2016). Interestingly, accumulation of cholesterol, sphingosine, sphingomyelin and glycosphingolipids in the lysosomes are observed both in Niemann-Pick and ASM deficiency cells and tissues (Leventhal et al., 2001; Vanier, 1983).

The NPC1 knock-out mouse (NPC1 KO) and a knock-in of the most common NPC1 patient mutation I1061T (Praggastis et al., 2015) are established models of Niemann-Pick type C disease (Loftus et al., 1997). Both NPC1 KO and NPC1$^{I1061T}$ mice recapitulate most of the neuropathological phenotypes of the disease, with the disease onset occurring earlier in the NPC1 KO mice. The ASM knock-out mouse (ASM KO) is a widely used model of ASM deficiency (Horinouchi et al., 1995).

Mitochondria are fundamental metabolic organelles in the cell, harboring key pathways for aerobic metabolism such as the citrate cycle, the key integrator metabolic pathway, as well as the respiratory chain and oxidative phosphorylation, Fe-S cluster and heme synthesis (Pagliarini and Rutter, 2013). They are also recognized as a major cellular signaling platform, with far-reaching implications on cell proliferation, stem cell maintenance, cellular immunity and cell death (Kasahara and Scorrano, 2014; Raimundo, 2014). Mitochondria are composed of about 1000 proteins, of which only 13 are encoded by mitochondrial DNA (mtDNA) (Pagliarini et al., 2008). The other ~1000 proteins are encoded by nuclear genes, and imported to the different sub-mitochondrial compartments (e.g., matrix, inner membrane, outer membrane, intermembrane space) by dedicated pathways (Wiedemann and Pfanner, 2017).

The large number of proteins that are nuclear-encoded and imported to mitochondria imply the need for regulatory steps that ensure the coordination of the process of mitochondrial biogenesis. This is often regulated at transcript level, by transcription factors that promote the expression of nuclear genes encoding for mitochondrial proteins (Scarpulla et al., 2012). One of the best characterized is the nuclear respiratory factor 1 (NRF1), which stimulates the expression of many subunits of the respiratory chain and oxidative phosphorylation, and also of genes necessary for mtDNA maintenance and expression, such as TFAM (Evans and Scarpulla, 1989; Evans and Scarpulla, 1990). Other transcription factors, such as estrongen-related receptor α (ERRα) and the oncogene myc, also act as positive regulators of mitochondrial biogenesis (Herzog et al., 2006; Li et al., 2005). Several co-activators also participate in the regulation of mitochondrial biogenesis, of which the co-activator PGC1α (peroxisome proliferator-activated receptor-gamma, co-activator 1 α) is the best characterized (Wu et al., 1999). PGC1a can interact with NRF1 or ERRα and stimulate mitochondrial biogenesis (Scarpulla et al., 2012). No transcriptional repressors of mitochondrial biogenesis have so far been described. Impaired or uncoordinated mitochondrial biogenesis often results in impaired mitochondria leading to pathological consequences (Cotney et al., 2009; Raimundo et al., 2012).

Here, we identify the transcription factors KLF2 and ETV1 as transcriptional repressors of mitochondrial biogenesis. The up-regulation of these two proteins in patient cells and mouse tissues of two lysosomal diseases, Niemann-Pick type C and ASM deficiency, underlies the mitochondrial defects observed in these syndromes. The silencing of ETV1 and, particularly, KLF2, is sufficient to return mitochondrial biogenesis and function to control levels.

## Results

### Expression of mitochondria-related genes is decreased in NPC1 KO tissues

Mitochondrial homeostasis and function is impaired in many lysosomal storage diseases. The two main axes of mitochondrial homeostasis are biogenesis and demise (by selective autophagy,

designated mitophagy). Given that lysosomal diseases are characterized by impaired autophagy (*Settembre et al., 2008*), it is expectable that mitophagy is also impaired. However, it remains unknown how mitochondrial biogenesis is affected in lysosomal storage diseases.

To assess mitochondrial biogenesis at transcript level in a systematic manner, we resorted to a publicly-available transcriptome dataset of NPC1 KO mice liver and brain, the two tissues most affected in Niemann-Pick type C. The dataset included both pre-symptomatic and symptomatic animals (*Alam et al., 2012*). To monitor the effects of Niemann-Pick disease on transcriptional regulation of mitochondrial biogenesis, we started by establishing a comprehensive list of mitochondria-related genes. We used a published mitochondrial proteome (MitoCarta, (*Pagliarini et al., 2008*) see Materials and methods for details), and converted the protein names to the corresponding ENSEMBL gene name to generate the 'mitochondria-associated gene list'. The process is illustrated in *Figure 1A*. We prepared a second list which included only the respiratory chain and oxidative phosphorylation subunits ('RC/OXPHOS gene list'). As controls, we prepared 'gene lists' for lysosomes, peroxisomes, Golgi and endoplasmic reticulum using the same strategy. The proteomes used to build the organelle-specific gene lists are detailed in the methods section (*Table 1*). Next, we used transcriptome data from asymptomatic and symptomatic brain and liver of NPC1 KO and corresponding WT littermates to determine how the organelle gene lists were affected.

First, we assessed the average expression of lysosomal genes in NPC1 KO brain and liver, to verify the validity of our 'organelle gene list' approach in this dataset. We have shown earlier that the average expression level of an organelle-gene list is a good indicator of the activity of the transcriptional program of biogenesis for that organelle (*Fernández-Mosquera et al., 2017*). The average expression of lysosomal genes was significantly increased in the asymptomatic NPC1 KO brain and liver (*Figure 1—figure supplement 1A*), and increased further with the onset of the disease in NPC1 KO brain and liver (*Figure 1—figure supplement 1A*), in agreement with the expected increase in the expression of lysosomal genes in lysosomal storage diseases.

Then, we measured the average expression of the 'mitochondrial gene list' in NPC1 KO brain and liver. Mitochondria-associated genes were up-regulated in pre-symptomatic NPC1 KO brain, and down-regulated in symptomatic brain (*Figure 1B*). In the liver, the average expression of mitochondria-associated genes was not significantly changed in the pre-symptomatic group, but was robustly decreased in the symptomatic NPC1 KO mice (*Figure 1B*). When looking only at the 'RC/OXPHOS gene list', the pattern was similar but the magnitude of the changes was more robust (*Figure 1C*). These results are not due to a small number of genes skewing the whole population, since the proportion of mitochondrial genes in the differentially expressed gene lists for NPC1 KO brain (*Figure 1—figure supplement 2A–C*) and liver (*Figure 1—figure supplement 2D–F*) increases robustly (about 5-fold) with disease onset. These results highlight a general trend towards a global down-regulation of mitochondrial genes under chronic lysosomal malfunction.

In order to determine if this effect was specific to mitochondria or also observed in other organelles, we tested how the average expression of peroxisomal-, endoplasmic reticulum- and Golgi-specific genes was affected. The expression of peroxisomal genes was not affected in NPC1 KO brain, but was down-regulated in both asymptomatic and symptomatic NPC1 KO liver (*Figure 1—figure supplement 1B*). The expression of endoplasmic reticulum-related and Golgi-related genes was not significantly altered (*Figure 1—figure supplement 1B*). These results suggest that lysosomal stress caused by absence of *Npc1* in multiple tissues specifically affects the expression of mitochondrial genes, although disease onset also results in a liver-specific repression of peroxisomal genes.

## Mitochondrial biogenesis and function are impaired in NPC and ASM patient cells and tissues

To verify the results from the large-scale transcriptional analysis of NPC1 KO tissues, we tested the expression of several genes encoding for mitochondrial proteins in the livers of NPC1 KO mice. The genes tested encode for subunits of the respiratory chain complex I (*NDUFS3* and *ND6*), complex II (*SDHA*), complex III (*CYTB*) and complex IV (*COX5A*, *COX1*). *ND6*, *CYTB* and *COX1* are encoded by mtDNA, while all the others are nuclear-encoded. We observed a robust and consistent decrease in the transcript levels of mitochondria-related genes in the livers of NPC1 KO mice (*Figure 2A*) compared to their respective WT littermates. A similar reduction on the expression of mitochondria-associated genes was also observed in NPC patient fibroblasts (*Figure 2B*) whose lysosomal phenotype has already been characterized (*Park et al., 2003*).

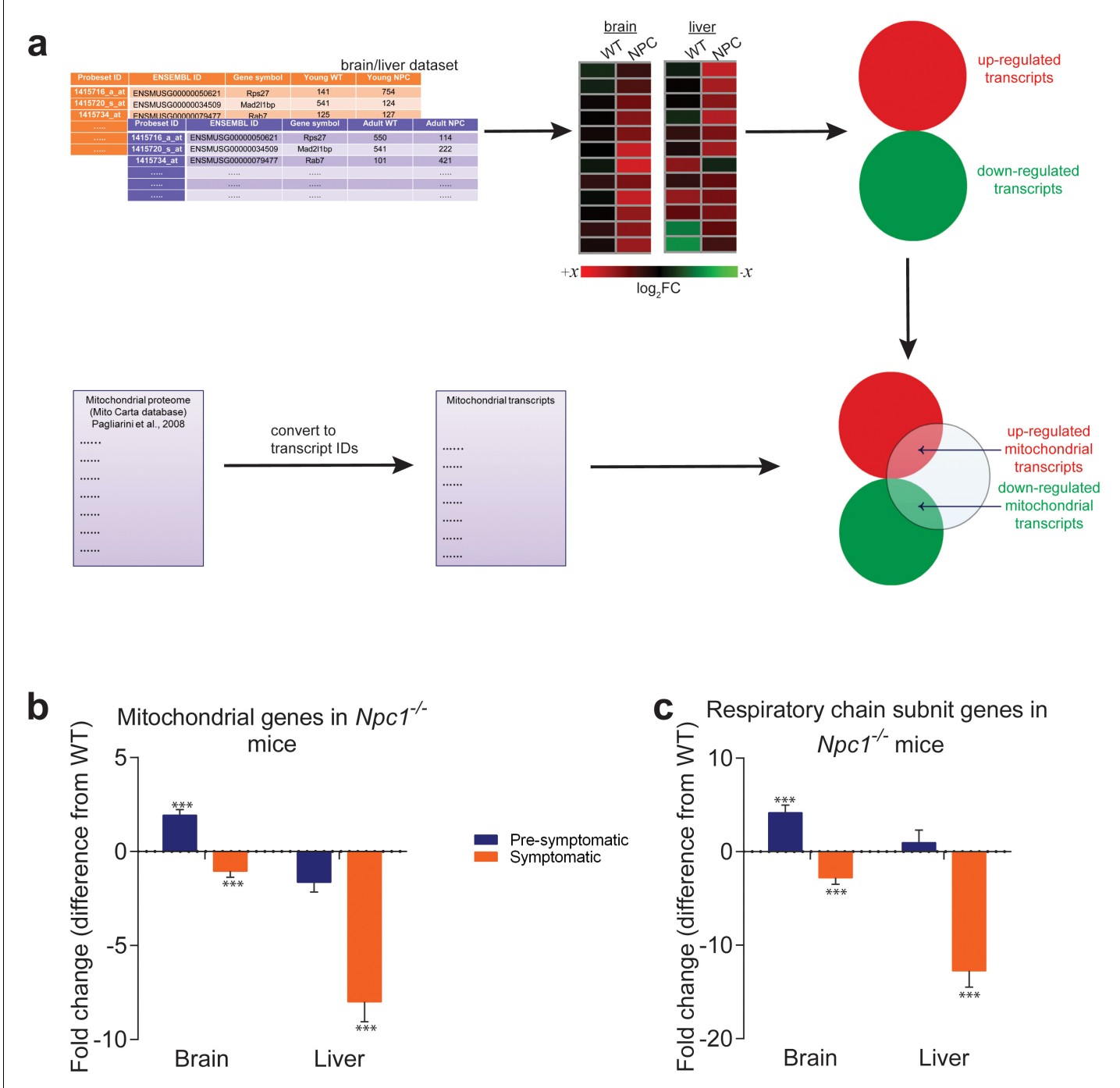

**Figure 1.** Mitochondrial genes are down-regulated in brain and liver of symptomatic NPC1 KO mice. (**a**) Schematic representation of the in silico approach. The list of mitochondria-related genes was built by converting the MitoCarta proteome inventory into a transcript list. This was then crossed with the differentially-expressed gene list of brain and liver in symptomatic and asymptomatic NPC1 KO (n = 11; brain, n = 6; liver) versus WT (n = 5; brain, n = 6; liver) mice. (**b–c**) Decreased expression of genes encoding ~1000 mitochondrial proteins (panel b) and ~100 respiratory chain subunits (panel c) in brain and liver of NPC1 KO mice. The plots (b–c) represent the variation in gene expression comparing the fold change between the average expression in NPC1 KO over NPC1 WT. Error bars denote standard error of the mean (s.e.m.). This variation is represented as the difference from average WT expression (e.g., a 20% increase in the mutant mice is shown as 20%, while −25% denotes a 25% decrease). Statistical analyses using t-test with Bonferroni correction, adjusted p-values ***p<0.001.

DOI: https://doi.org/10.7554/eLife.39598.003

The following figure supplements are available for figure 1:

*Figure 1 continued on next page*

*Figure 1 continued*

**Figure supplement 1.** Behaviour of organelle-specific gene lists in NPC1 KO tissues.
DOI: https://doi.org/10.7554/eLife.39598.004
**Figure supplement 2.** Disease progression changes in the expression of mitochondria-related genes in brain and liver of NPC1 KO mice.
DOI: https://doi.org/10.7554/eLife.39598.005

The accumulation of cholesterol and sphingomyelin in the lysosomes is common to both NPC and acid shingomyelinase (ASM) deficiency (*Pentchev et al., 1984*; *Reagan et al., 2000*; *Leventhal et al., 2001*; *Herzog et al., 2006*; *Lloyd-Evans et al., 2008*; *Suzuki et al., 2012*; *Skon et al., 2013*; *Platt, 2014*). However, while mitochondria in NPC also present increased levels of cholesterol, this does not happen in ASM deficiency (*Torres et al., 2017*). Since excessive mito-chondrial cholesterol can impair mitochondrial function (*Torres et al., 2017*), we tested if ASM defi-ciency would also have a repressive effect on mitochondrial biogenesis. Similar to the NPC findings, we observed a decrease in the expression of mitochondria-associated genes in the ASM KO liver compared to the WT littermates (*Figure 2C*) as well as in two different patient fibroblasts of ASM deficiency (*Figure 2D*).

To assess if this down-regulation of mitochondrial biogenesis in NPC and ASM deficiency had functional consequences for respiratory chain efficiency, we measured the amounts of mitochondrial superoxide, a by-product of the mitochondrial respiratory chain known to be produced in higher amounts when mitochondria are not functioning optimally (*Raimundo et al., 2012*; *Rai-mundo, 2014*), which can be estimated using a superoxide-sensitive mitochondria-targeted dye, MitoSox. We observed an increase in MitoSox intensity in patient fibroblasts with NPC (*Figure 2E*) and ASM deficiency (*Figure 2F*) denoting increased superoxide levels which are indicative of poor mitochondrial performance. Altogether, these results show that the biogenesis of mitochondria is repressed in NPC- and ASM-deficient cells and tissues, and that the existing mitochondria are not functioning optimally. Furthermore, the mitochondrial impairments are likely unrelated to the levels of cholesterol in mitochondria (known to be high in NPC but normal in ASM; *Torres et al., 2017*), and seem rather a consequence of the lysosomal saturation in NPC and ASM deficiency.

## Impaired mitochondrial respiration in NPC1 and ASM deficiency

To further characterize the impact of lysosomal disease on mitochondrial function, we focused on the ASM-deficient fibroblasts, which showed a more robust decrease of mitochondrial biogenesis than NPC and do not have the confounding factor of excessive mitochondrial cholesterol. We used cells from two patients of ASM deficiency, one of which (ASM-2) had the lysosomal phenotype already characterized (*Corcelle-Termeau et al., 2016*). Additionally, we also employed a line from a patient (ASM-1) with compound heterozygous loss-of-function mutations in SMPD1 (the gene encod-ing ASM), which has severe ASM deficiency (5% activity left). The lysosomal impairments in this line have not yet been characterized besides patient diagnosis; therefore, we first evaluated lysosomal function in these fibroblasts. One of the consequences of lysosomal dysfunction is the accumulation of autophagic substrates, such as the protein p62 (also known as Sequestosome 1, SQSTM1) as well as autophagosomes (*Settembre et al., 2008*). We assessed the levels of p62/SQSTM1 and LC3B-II, a marker of autophagosomal mass, by Western blot, and found both sharply increased in the ASM-1 fibroblasts, as expected (*Figure 3—figure supplement 1A*). We also assessed the lysosomal proteo-lytic capacity, by measuring the degradation of the lysosomal substrate DQ-BSA. DQ-BSA is a poly-mer of fluorescently-tagged bovine serum albumin, which accumulates in the lysosomes. The fluorescence is quenched in the polymeric form and detectable in the monomers. As the lysosomal proteases start cleaving DQ-BSA and releasing monomers, fluorescence starts increasing, and the rate of fluorescence increase is proportional to the activity of lysosomal proteases. We observed a strong decrease in DQ-BSA degradation rate in the ASM-1 fibroblasts (*Figure 3—figure supple-ment 1B*). These results support a strong impairment of lysosomal function in ASM-1 cells used in this study, in line with the cellular phenotype of the disease and the described phenotype of ASM-2.

We then set to characterize mitochondrial function. First, we monitored the oxygen consumption rate (OCR). This was done with a high-throughput real-time respirometer, which allows the measure-ment under multiple conditions, such as basal medium, inhibition of oxidative phosphorylation (when OCR is inhibited) and uncoupled respiratory chain (when OCR occurs unrestrained). We

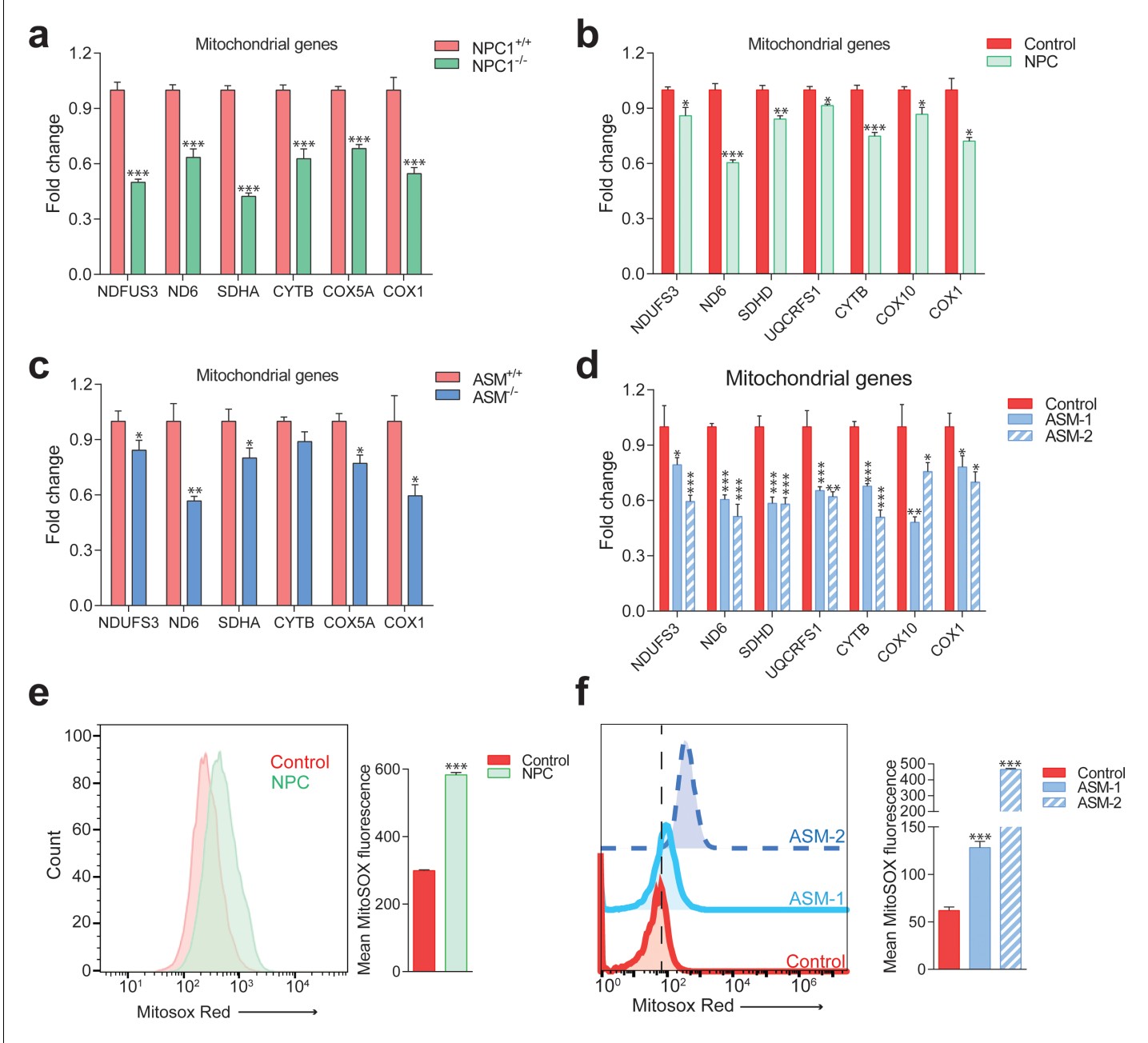

**Figure 2.** Impaired mitochondrial biogenesis and function in mouse and cellular models of Niemann-Pick disease. The transcript levels of several nuclear-encoded and mitochondrial DNA (mtDNA)-encoded mitochondria-related genes were measured. (**a**) transcript levels of mitochondria-related genes are decreased in the liver of NPC1 knockout mice (NPC1 KO), a model of Niemann-Pick type C. The plot shows mean ± s.e.m. T-test p-values ***p<0.001, n = 9 (**b**) transcript levels of mitochondria-related genes are decreased in the fibroblasts of a patient with compound heterozygote NPC1 mutations (GM18398 Coriell Repository). The plot shows mean ± s.e.m. T-test p-values *p<0.05 **p<0.01 ***p<0.001, n = 3 (**c**) transcript levels of mitochondria-related genes are decreased in the liver of acid sphingomyelinase knockout (ASM KO) mice, a model of acid sphingomyelinase deficiency. The plot shows mean ± s.e.m. T-test p-values *p<0.05 **p<0.01, n = 8. (**d**) transcript levels of mitochondria-related genes are decreased in fibroblasts from a patient with acid sphingomyelinase deficiency (only 5% of ASM activity left) and in the ASM-2 patient line. The plot shows mean ± s. e.m. T-test p-values *p<0.05 **p<0.01 ***p<0.001, n = 3. Further characterization of the lysosomal defects in the fibroblasts of this patient are presented in *Figure 3—figure supplement 1*. (**e–f**) mitochondrial superoxide levels, as assessed by the fluorescence intensity of the superoxide-sensitive mitochondria-targeted dye MitoSox, measured by flow cytometry, are increased in NPC fibroblasts (panel e) and in ASM-1 and ASM-2 patient fibroblasts (panel f); histogram plots are representative of three biological replicates. Quantifications denote mean ± s.e.m..T-test p-values ***p<0.001, n = 3.

DOI: https://doi.org/10.7554/eLife.39598.006

observed a robust decrease in OCR in ASM-1 fibroblasts which lasted across all conditions tested: basal medium, inhibition of the oxidative phosphorylation with oligomycin, and uncoupling of respiratory chain and oxidative phosphorylation by FCCP (*Figure 3A*). We determined that the ASM-1 fibroblasts have ~70% decrease in the OCR compared to the control cells in basal conditions and in

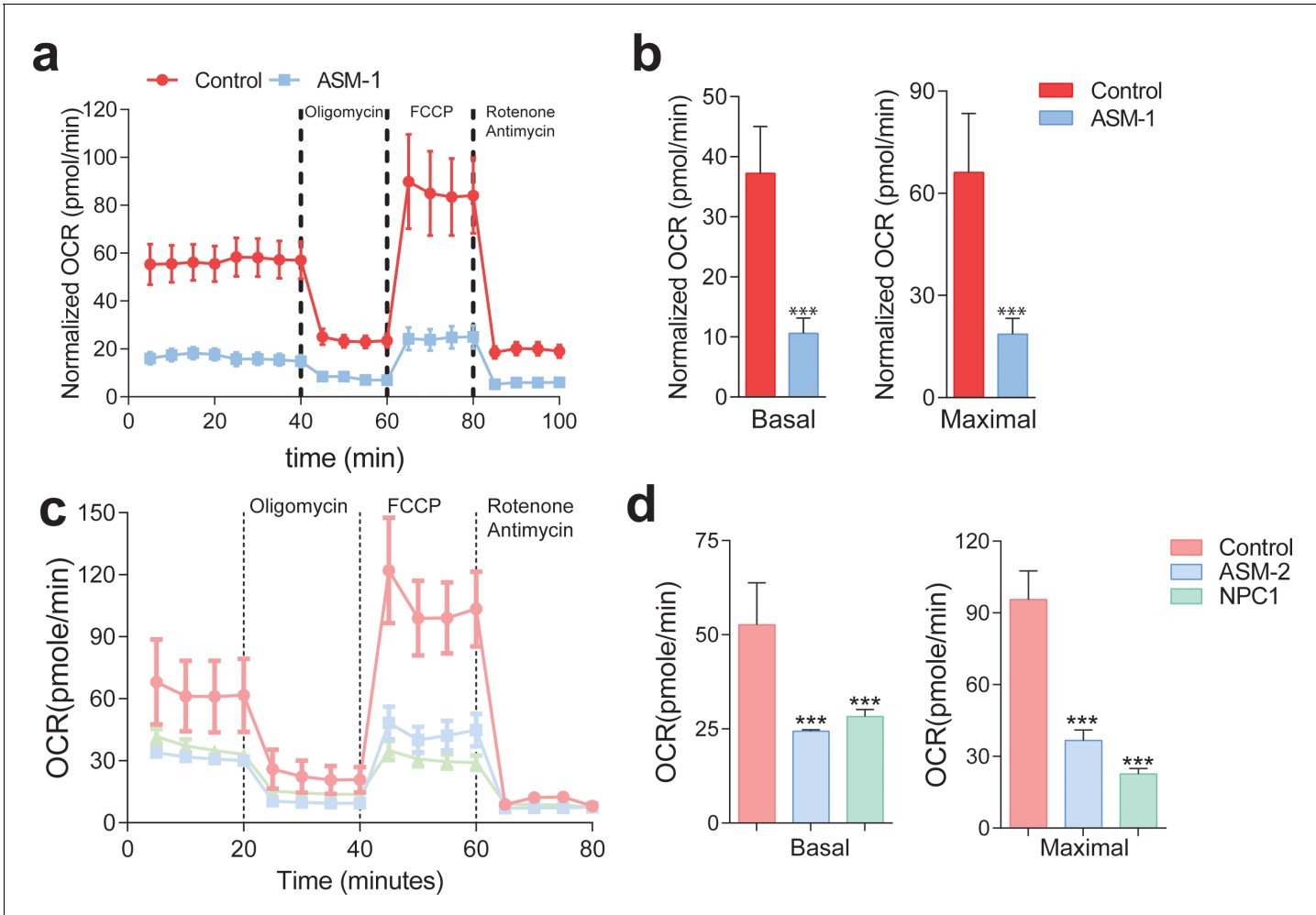

**Figure 3.** Mitochondrial function and mitochondrial mass are impaired in acid sphingomyelinase (ASM)- and NPC1-deficient patient fibroblasts. (a,c) ASM- and NPC1-deficient fibroblasts have substantially lower $O_2$ Consumption Rate (OCR) than controls. OCR was measured using whole cells, sequentially in basal conditions (complete medium), after oxidative phosphorylation inhibition using the ATPase inhibitor oligomycin, after uncouling the respiratory chain from oxidative phosphorylation using the uncoupler FCCP, and after inhibition of the respiratory chain using complex I inhibitor rotenone and complex I II inhibitor antimycin. The measurements were made in a 96-well plate using a SeaHorse Extracellular Flux analyser. The mean ± s.e.m. of at least eight wells per cell line is plotted over time. OCR was normalized to the amount of protein in each well. (b,d) Reduced basal and maximal (uncoupled) OCR in ASM1-deficient fibroblasts quantified from the curves in (a) and in ASM2- and NPC1- deficient fibroblasts quantified from profiles in (c) and bar graphs are presented as mean ± s.e.m. T-test p-value ***$p < 0.001$, n = 3.

DOI: https://doi.org/10.7554/eLife.39598.007

The following figure supplements are available for figure 3:

**Figure supplement 1.** Validation of lysosomal defects in patient fibroblasts and causal relationship of defects to mitochondrial biogenesis and function.

DOI: https://doi.org/10.7554/eLife.39598.008

**Figure supplement 2.** Increased content of dysfunctional mitochondria in ASM-deficient and NPC fibroblasts.

DOI: https://doi.org/10.7554/eLife.39598.009

**Figure supplement 3.** Mitochondrial deficits in control fibroblasts treated with desipramine 40 µM for 72 hr (inhibitor of acid sphingomyelinase).

DOI: https://doi.org/10.7554/eLife.39598.010

**Figure supplement 4.** Mitochondrial deficits in control fibroblasts treated with U18666A 10 µM for 72 hr (inhibitor of NPC1).

DOI: https://doi.org/10.7554/eLife.39598.011

maximal (uncoupled) conditions (*Figure 3B*). We also monitored the OCR in ASM-2 and NPC fibroblasts (*Figure 3C*) and observed that they also presented a robust decrease in OCR both at basal (~50% down) and maximal conditions (*Figure 3D*). Importantly, the expression of wild-type ASM in ASM-1 and ASM-2 fibroblasts increased OCR significantly, as did expression of wild-type NPC1 in NPC1-deficient fibroblasts (*Figure 3—figure supplement 1C–D*). Furthermore, introduction of wild-type ASM relieved the inhibition in the expression of mitochondrial genes, denoted by the increase in their transcript levels (*Figure 3—figure supplement 1E*). Expression of wild-type NPC1 in NPC1-deficient fibroblasts had a similar result (*Figure 3—figure supplement 1E*). These results show that the robust decrease in mitochondrial biogenesis and respiration observed in ASM- and NPC1-deficient fibroblasts are a specific consequence of the loss of ASM or NPC1 activity, respectively.

We then tested if the decrease in mitochondrial biogenesis in ASM- and NPC1-deficient fibroblasts resulted in decreased mitochondrial mass. We stained the cells with a dye that specifically targets mitochondria independently of the mitochondrial inner membrane potential, Mitotracker green, and measured the intensity of this dye in the different lines. We observed that there was increased Mitotracker green signal in the three lines with lysosomal defects (*Figure 3—figure supplement 2A*), suggesting increased mitochondrial mass. We then assessed the protein levels of respiratory chain proteins. Despite their transcripts were repressed, we did not observe a similar reduction in the protein levels (*Figure 3—figure supplement 2B*). These results suggest that despite the transcriptional repression of mitochondrial biogenesis, there is accumulation of mitochondria, likely due to the impact of impaired lysosomal function on mitophagy. This would result in the accumulation of damaged (e.g. uncoupled) mitochondria in the cytoplasm, which would normally be removed in cells with functioning lysosomes/autophagy pathway. Thus, we stained the cells with two dyes that accumulate specifically in mitochondria, Mitotracker green (independent of mitochondrial inner membrane potential) and Mitotracker red (potential-dependent mitochondrial accumulation), and analyzed the intensity of the signals by flow cytometry. We quantified the proportion of cells with low-potential mitochondria (essentially, a simultaneous decrease in the red intensity, and increase in the green intensity). This proportion is increased in all lysosomal patient cells tested (*Figure 3—figure supplement 2C*). Control cells treated with the mitochondrial uncoupler CCCP were used as positive control for this assay, and show a massive increase in the proportion of cells with uncoupled mitochondria. Therefore, the ASM- and NPC1-deficient cells present accumulation of dysfunctional mitochondria in the cytoplasm, despite the repression in mitochondrial biogenesis at transcript level. These results are further underscored by the decrease in mitochondrial respiratory activity and the increase in superoxide levels in the ASM- and NPC1-deficient cells, as shown above (*Figure 2E–F* and *Figure 3*).

Acid sphingomyelinase generates ceramide, which is itself a powerful signaling lipid, and can be metabolized by acid ceramidase into sphingosine and other signaling lipids. Notably, desipramine inhibits both acid sphingomyelinase and acid ceramidase (*Herzog et al., 2006*). We then tested if simultaneous pharmacological blockage of ASM and acid ceramidase with desipramine yields the same mitochondrial phenotypes as observed in the ASM patient cells. In the desipramine-treated cells we observed decreased mitochondrial biogenesis (lower transcript levels of genes encoding mitochondrial proteins, *Figure 3—figure supplement 3A*) and accumulation of damaged mitochondria, as shown by increased superoxide levels (*Figure 3—figure supplement 3B*) and decreased respiration (*Figure 3—figure supplement 3C–D*). Thus, pharmacological inhibition of both ASM and acid ceramidase yielded similar results to ASM patient cells, suggesting that acid ceramidase is not relevant for the phenotypes observed.

Since one of the known consequences of ASM deficiency is accumulation of cholesterol in the lysosomes (*Lloyd-Evans et al., 2008*; *Yeo et al., 2014*), and given that we observed similar perturbations on mitochondrial homeostasis in ASM- and NPC1-deficient patient fibroblasts, we tested if pharmacological inhibition of NPC1 would also be sufficient to impact mitochondrial biogenesis and function. We treated control cells with the NPC1 inhibitor U18666A (*West et al., 2015*), and observed decreased expression of mitochondria-associated genes (*Figure 3—figure supplement 4A*), and increased mitochondrial superoxide levels (*Figure 3—figure supplement 4B*). Finally, treatment with U18666A resulted in lower respiration with ~30% lower basal OCR and ~50% lower uncoupled OCR (*Figure 3—figure supplement 4C–D*). Thus, pharmacological inhibition of ASM or NPC1, similar to genetic defects in these proteins, is sufficient to cause decreased expression of mitochondrial genes and impaired mitochondrial respiratory chain activity.

## KLF2 and ETV1 are up-regulated in NPC1 KO tissues and repress transcription of mitochondria-associated genes

Having established a clear mitochondrial phenotype in NPC1 and ASM deficiency, we set out to identify the underlying mechanism. The robust decrease in the expression of hundreds of mitochondria-related genes in NPC1 KO brain (*Figure 1—figure supplement 2B–C*) and NPC1 KO liver (*Figure 1—figure supplement 2E–F*) suggests the involvement of a coordinated transcriptional program, and therefore of transcriptional regulators such as transcription factors. To determine which transcription factors might be mediating the repression of mitochondria-associated genes, we took an unbiased bottom-up approach to determine potential transcriptional regulators. Given that the whole mitochondrial gene list has ~1000 genes, we focused on the RC/OXPHOS list, which shows the same behavior as the complete mitochondrial gene list (as shown in *Figure 1*) and has a more manageable size (~100 genes). Using the Genomatix Gene2Promoter tool, we obtained the genomic sequences (Mus musculus) of the promoter regions of the RC/OXPHOS genes, from −500 base pairs upstream the transcription start site, to +100 base pairs downstream. This region is sufficient to account for the regulation of gene expression by transcription factors in many promoters of mitochondrial genes (*Gleyzer et al., 2005*; *Virbasius and Scarpulla, 1994*). We then used Genomatix Matinspector tool to analyze the gene promoters for transcription factor binding sites (cis-elements), and identified those statistically enriched (illustrated in *Figure 4A*). The most overrepresented cis-elements in the promoters of RC/OXPHOS genes were the transcription factor families SP1, E2F, Krueppel-like factors (KLF) and ETS factors (*Table 2*). In parallel, as control, we carried out a similar approach for the lysosomal gene list (whose expression is increased, in contrast to the mitochondrial genes) and observed that the SP1 and E2F families were also significantly enriched in the promoters of lysosomal genes (*Supplementary file 1*). Given that the expression of lysosomal genes and mitochondrial genes is affected in opposite ways, we reasoned that it would be unlikely that the same transcription factors were driving two opposite processes. For this reason, we proceeded only with the KLF and ETS families, which only scored as significantly enriched in the mitochondrial promoters (*Table 2*).

Next, we again resorted to the transcriptome dataset of NPC1 KO brain and liver to determine if any transcription factors in the KLF2 and ETS families were predicted to have increased or decreased activity during NPC disease progression. Using Ingenuity Pathway Analysis, we determined which transcription factors scored as significant regulators in these tissues (*Supplementary file 2*). The only transcription factor of the KLF family meeting the criteria was KLF2. Several ETS family transcription factors have redundant binding sites (*Hollenhorst et al., 2007*), so we tested the three members that scored in the Genomatix promoter analysis, SPI1, ELK1 and ETV1. SPI1 is expressed in macrophages and not expressed in fibroblasts (*Feng et al., 2008*; *Suzuki et al., 2012*), and accordingly we could not detect the expression of *SPI1* in control or patient fibroblasts, either at transcript or protein (data not shown). While *ELK1* was not changed at transcript level (*Figure 4—figure supplement 1A*), *ETV1* was significantly increased in ASM deficiency patient fibroblasts (*Figure 4—figure supplement 1A*). The transcript levels of *KLF2* were not changed in ASM deficiency (*Figure 4—figure supplement 1B*).

We then focused on KLF2 and ETV1 (*Figure 4A*). First, we tested if the levels of these proteins were affected in NPC1- or ASM-deficient fibroblasts, by Western blotting. We found that both KLF2 and ETV1 were robustly up-regulated in both ASM patient lines (*Figure 4B*). In the NPC1- deficient cells, KLF2 was robustly increased, but ETV1 was not significantly changed (*Figure 4B*). Given that many transcription factors shuttle between the nucleus and the cytoplasm, we prepared nuclear extracts to verify if KLF2 and ETV1 were enriched in the nucleus of the patient cells. We observed that there was a clear increase in nuclear KLF2 and ETV1 in ASM-1 cells (*Figure 4C*). Similarly, ASM-2 and NPC1-deficient cells also had increased nuclear KLF2 (*Figure 4—figure supplement 1C*) and ETV1 (*Figure 4—figure supplement 1D*). Thus, KLF2 and ETV1 are likely more active both in ASM- and NPC1-deficient cells.

We again compared the ASM-deficient fibroblasts with control fibroblasts treated with the inhibitor of both ASM and acid ceramidase. Desipramine-treated fibroblasts yielded a similar result: both KLF2 and ETV1 are up-regulated at protein level (*Figure 4—figure supplement 2A*, quantified in *Figure 4—figure supplement 2B*) but only ETV1 transcript levels are significantly changed (*Figure 4—figure supplement 2C*). Altogether, these results suggest that the accumulation of KLF2 in

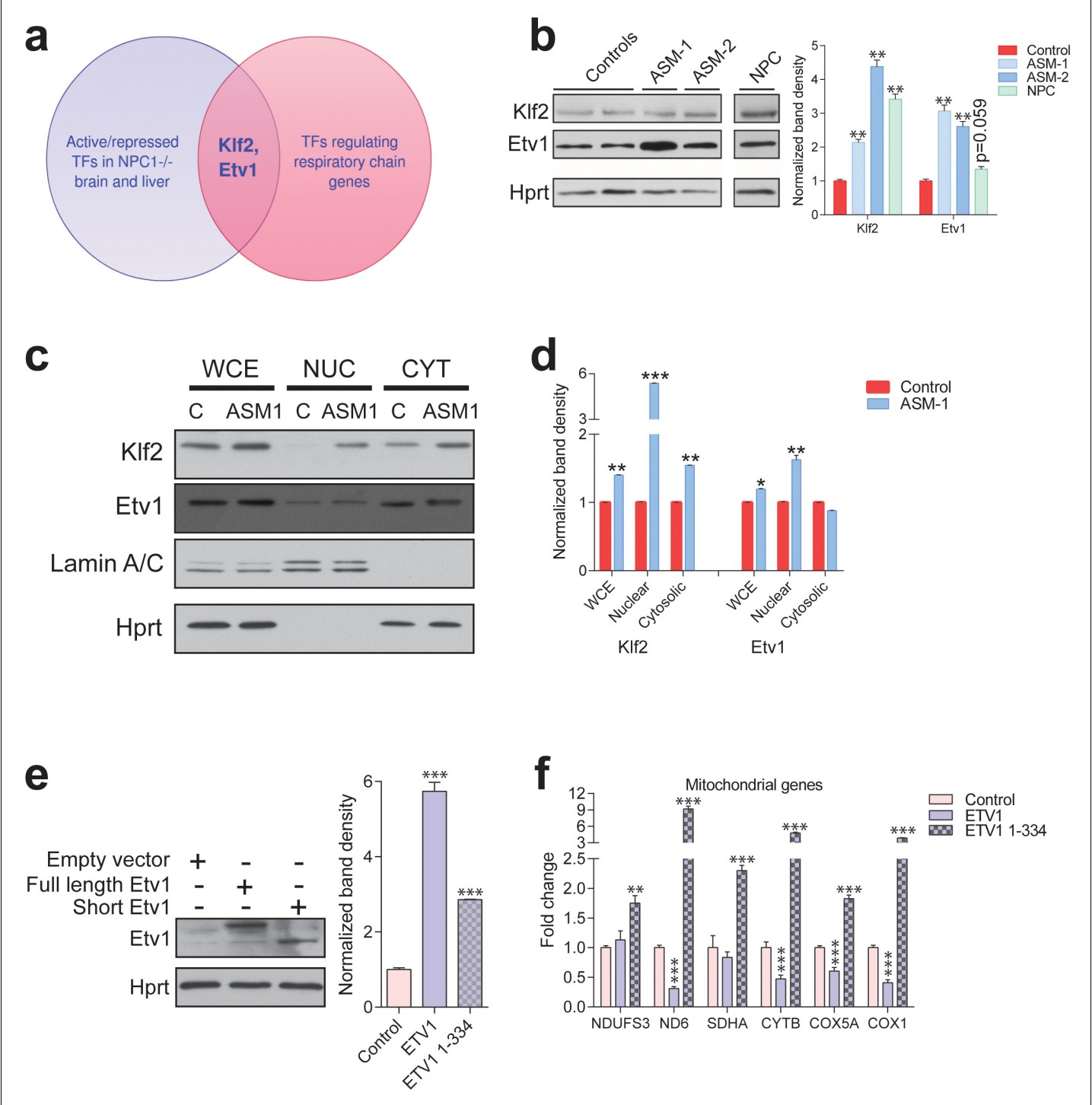

**Figure 4.** Transcription factors Etv1 and Klf2 are induced in Niemann-Pick and involved in the regulation of mitochondrial biogenesis. (**a**) Venn diagram illustrating the intersection between the list of transcription factors (TFs) that are significantly activated or repressed in tissues of *Npc1^-/-* mice, and the list of TFs that are predicted to regulate the expression of mitochondrial respiratory chain genes, which yields KLF2 and ETV1 as hits. (**b**) Increased Klf2 and Etv1 protein levels in ASM-deficient and NPC fibroblasts, shown in a representative (out of three biologically independent experiments) western blot of whole cell extracts, with quantification (mean ± s.e.m., n = 3) of band densities in the adjacent plot. T-test p-values **p<0.01 (**c**) Increased nuclear localization of Klf2 and Etv1 in ASM-deficient fibroblasts. Blots are representative of biological triplicates with quantifications in (**d**) shown as mean ± s.e.m. T-test p-values *p<0.05, **p<0.01 and ***p<0.001 (**e**) Overexpression of ETV1^WT (full length ETV1) and of ETV1^1-334, lacking the C-terminus which includes the DNA-binding domain. Representative western blot, quantification of band densities normalized to empty vector control from two independent experiments with two technical replicates each on the right panel (mean ± s.e.m.) (**f**) Overexpression of ETV1^WT significantly

*Figure 4 continued on next page*

*Figure 4 continued*

down-regulates the transcript levels of most mitochondria-related genes, while ETV1[1-334], unable to bind DNA, causes an increase in transcript levels. The plots show mean ± s.e.m. T-test p-values **p<0.01 ***p<0.001, n = 2 with three technical replicates each.

DOI: https://doi.org/10.7554/eLife.39598.013

The following figure supplements are available for figure 4:

**Figure supplement 1.** Etv1 and Klf2 levels in ASM-deficient and NPC fibroblasts.

DOI: https://doi.org/10.7554/eLife.39598.014

**Figure supplement 2.** Etv1 and Klf2 levels in desipramine-treated fibroblasts.

DOI: https://doi.org/10.7554/eLife.39598.015

**Figure supplement 3.** Increased expression of mitochondria-related genes in the absence of KLF2.

DOI: https://doi.org/10.7554/eLife.39598.016

**Figure supplement 4.** Targets of KLF2 as determined by ChIP-Seq data analysis.

DOI: https://doi.org/10.7554/eLife.39598.017

**Figure supplement 5.** Targets of ETV1 as determined by ChIP-ChIP data analysis.

DOI: https://doi.org/10.7554/eLife.39598.018

response to lysosomal lipid storage is regulated post-translationally, while ETV1 is regulated at transcript level. The nuclear localization of both transcription factors is likely another regulatory step for KLF2 and ETV1 in ASM- and NPC1-deficient cells.

Given that ETV1 and KLF2 are predicted by our promoter analysis to have binding sites in the promoters of the genes encoding for respiratory chain subunits, and that increased expression of these two transcription factors correlates with repression of respiratory chain genes, we reasoned that KLF2 and ETV1 might be mediating this repression. To explore this possibility, we took advantage of another publicly available transcriptome dataset of erythroid cells of KLF2 KO and WT mice (GSE27602) (*Redmond et al., 2011*). We observed an increase in the average transcript levels of the 'mitochondria gene list' in the KLF2 KO cells compared to the WT littermates (*Figure 4—figure supplement 3*). The effect is also observed, with higher magnitude, when measuring the average expression of the genes encoding for respiratory chain subunits (*Figure 4—figure supplement 3*). These results suggest that KLF2 is able to repress mitochondrial biogenesis in vivo. This effect is likely direct, since analysis of a KLF2 ChIP-Seq dataset (*Yeo et al., 2014*) reveals a large number of target genes encoding mitochondrial proteins, including several respiratory chain subunits (*Figure 4—figure supplement 4*), in agreement with our in silico promoter analysis. Notably, ETV1 and other transcription factors regulating mitochondrial biogenesis, such as NRF1, were also identified as KLF2 transcriptional targets in the same dataset (*Figure 4—figure supplement 4*).

In addition, it is noteworthy that several known ETV1 targets are mitochondrial genes, as previously shown by chromatin immunoprecipitation (*Baena et al., 2013*) and illustrated in *Figure 4—figure supplement 5*. To test if the effect of ETV1 on the expression of mitochondria-related genes is direct, we expressed full length ETV1 (ETV1[FL]) as well as ETV1 lacking the DNA-binding domain (ETV1[1-334]) in control fibroblasts (*Figure 4E*) and evaluated the effect on the expression of mitochondria-related genes. The overexpression of ETV1[FL] elicited a decrease in the transcript levels of most mitochondria-associated genes (*Figure 4F*). However, ETV1[1-334] did not repress the transcript levels of these mitochondrial-related genes (*Figure 4F*). This result is coherent with the role of ETV1 as a repressor of mitochondrial biogenesis, and further demonstrates that this repression occurs via direct binding of ETV1 to DNA (*Janknecht, 1996*), thus validating our in silico promoter analysis. The unexpected increase in the transcript levels of mitochondria-related genes under overexpression of ETV1[1-334], unable to bind DNA, may be explained by ETV1 functioning as a homodimer (*Poon, 2012*). Therefore, overexpression of a mutant unable to bind DNA might titrate out the wild-type ETV1, thus effectively functioning as a dominant-negative ETV1 isoform, with the consequent activation of mitochondrial biogenesis.

## Silencing of KLF2 and ETV1 in ASM- and NPC1-deficiency rescues mitochondrial biogenesis and function

To test if KLF2 and ETV1 were indeed repressing mitochondrial biogenesis in ASM-deficient cells, we knocked-down ETV1 (*Figure 5A*) and KLF2 (*Figure 5B*), independently, in ASM-deficient fibroblasts. Given that the ASM-1 and ASM-2 fibroblasts had the same mitochondrial phenotype, and

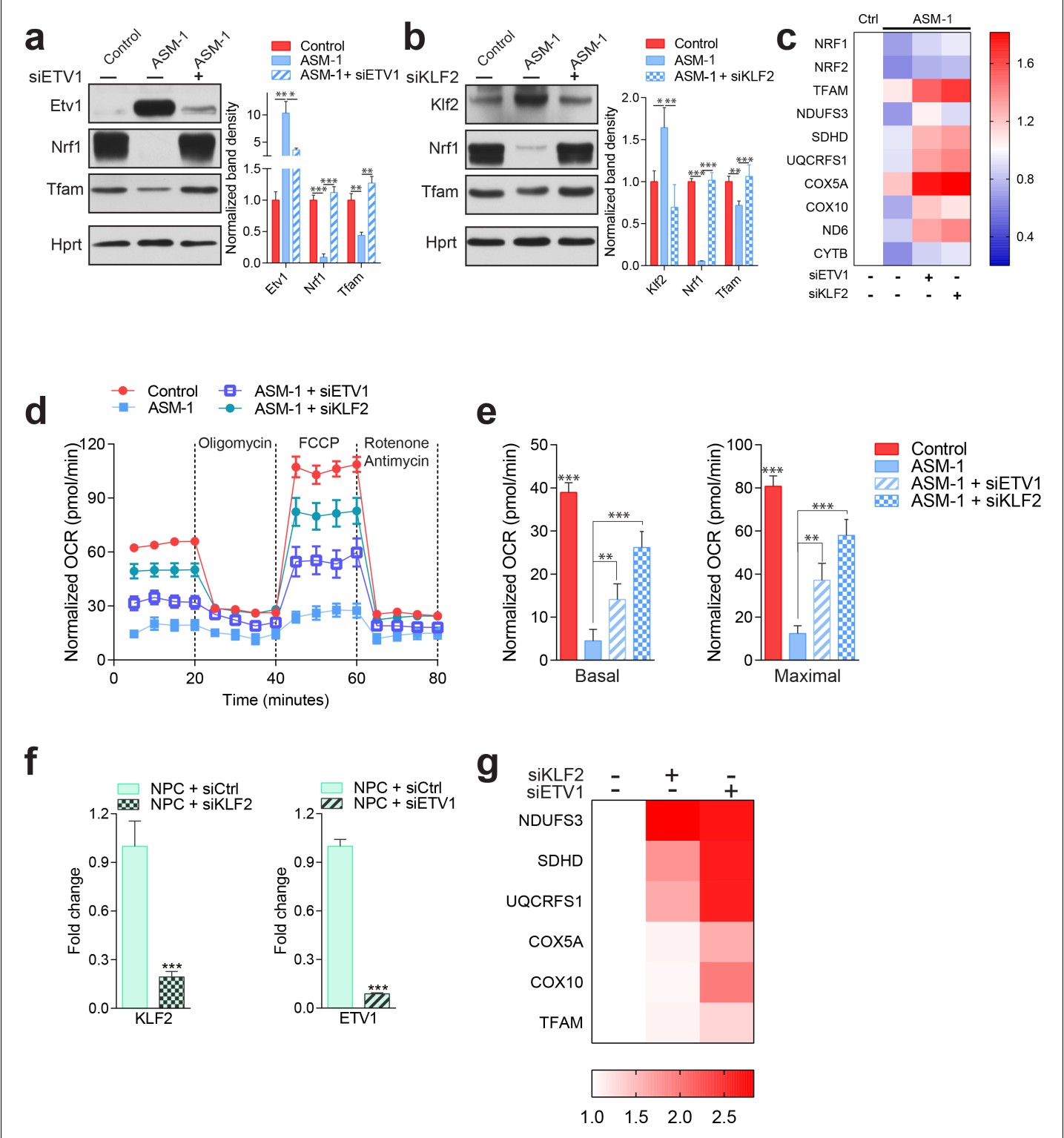

**Figure 5.** Silencing of ETV1 or KLF2 rescues mitochondrial biogenesis and function in Niemann-Pick fibroblasts. Using siRNA-mediated silencing, we knocked-down Etv1 (a) or Klf2 (b) in ASM1-deficient fibroblasts, which brought the protein levels of mitochondrial protein TFAM, and of mitochondrial biogenesis regulator NRF1 to control levels, as shown in a representative western blot of whole cell extracts, with quantification of band densities in the adjacent plots as mean ± s.em., n = 3. Scrambled siRNA was used as control in both control and ASM-deficient cells for all experiments involving ETV1 or KLF2 silencing. T-test p-values *p<0.05, **p<0.01 ***p<0.001 (c) Silencing of ETV1 or KLF2 increases the transcript levels of mitochondrial genes, as assessed by qPCR. The data is presented in a Heatmap, in which blue denotes decrease in expression compared to the control cells (white represents
*Figure 5 continued on next page*

Figure 5 continued

no change relative to the control values) and red denotes increase. Note the mostly decreased (blue) mitochondrial genes in ASM-deficient cells and their turn to red (increased expression) when ETV1 or KLF2 are silenced (n = 3). (d–e) Silencing of either Klf2 or Etv1 partially rescues the decreased basal and maximal OCR in ASM-deficient fibroblasts as measured by real time respirometry. The plot shows the mean ± s.e.m., n = 3. T-test p-values **p<0.01 and ***p<0.001. (f) Robust silencing of KLF2 or ETV1 in NPC1-deficient cells shows accordingly, significantly decreased transcript levels of *KLF2* and *ETV1*. Graphs represent mean ± s.e.m, n = 3 with T-test p-values ***p<0.001 (g) KLF2 or ETV1 knockdowns in NPC1-deficient cells increases the transcript levels of mitochondrial genes, which is presented as a Heatmap. Note the mostly increased (red) mitochondrial genes when KLF2 or ETV1 are silenced relative to Scrambled siRNA (white) in NPC1-deficient cells (n = 3).

DOI: https://doi.org/10.7554/eLife.39598.019

The following figure supplement is available for figure 5:

**Figure supplement 1.** Autophagy defects in Niemann Pick patients are independent of Klf2 and Etv1.

DOI: https://doi.org/10.7554/eLife.39598.020

showed similar patterns of KLF2 and ETV1 behavior, at this point we focused on ASM-1, which had a slightly more robust effect.

The knock-downs of ETV1 and KLF2 were both effective (*Figure 5A–B*). Interestingly, transcription factor nuclear respiratory factor 1 (NRF1), a known inducer of mitochondria-related gene expression, was also sharply down-regulated in ASM-deficient fibroblasts, and was rescued by the silencing of ETV1 or of KLF2. This result suggests a compound effect of repression of mitochondria-related genes by KLF2 and ETV1, combined with decreased activation of the expression of the same genes by NRF1. We have shown above (*Figure 3—figure supplement 2*) that the ASM-deficient cells accumulate mitochondria, and for that reason show increased levels of mitochondrial proteins. Nevertheless, some mitochondrial proteins are present at lower levels in ASM1-deficient cells, of which TFAM is a notable example. TFAM is a target of NRF1, and its protein levels are sharply decreased in ASM-1 cells, but are readily normalized by silencing of ETV1 (*Figure 5A*) or KLF2 (*Figure 5B*). Importantly, the transcript levels of genes encoding mitochondrial proteins, which are down-regulated in ASM-deficient fibroblasts, were increased by the silencing of ETV1 and even more robustly increased by KLF2 silencing (*Figure 5C*). This pattern also includes NRF1 and its closely related protein nuclear respiratory factor 2 (NRF2, also known as GABPA), again suggesting that these two transcription factors may be repressed by KLF2 and ETV1. Importantly, the improvement in the expression of mitochondria-associated genes by silencing KLF2 or ETV1 is not due to an improvement of the lysosomal phenotype. We measured readouts of lysosomal function such as the accumulation of autophagosomal marker LC3BII or autophagy substrate p62, by Western blot, and found that silencing of KLF2 or ETV1 had no impact on the lysosomal dysfunction in ASM-deficient cells (*Figure 5—figure supplement 1*). Finally, mitochondrial respiration was partly rescued in ASM-1 fibroblasts by the knock-down of ETV1 and robustly rescued by KLF2 silencing (*Figure 5D*), both under basal and maximal electron flow conditions (*Figure 5E*).

To ensure that the effect of KLF2 and ETV1 on mitochondria is not limited to ASM-deficient cells, we also tested how the silencing of KLF2 and ETV1 impacts NPC1-deficient cells. The knock-downs were robust (*Figure 5F*), and resulted in a strong increase in the expression of mitochondria-related genes (*Figure 5G*).

Altogether, these results show that KLF2 and ETV1, two transcription factors that are increased in ASM- and NPC1-deficient fibroblasts and hyperactive in NPC1 KO tissues, repress mitochondrial biogenesis and that their silencing restores mitochondrial biogenesis and function in ASM- and NPC1-deficient fibroblasts.

## KLF2 regulates ETV1 in an ERK-dependent manner

The silencing of KLF2 had a more robust effect on the recovery of mitochondrial function than the silencing of ETV1. For this reason, we set to understand if these transcription factors work in parallel pathways or if they are epistatic. We observed that the silencing of KLF2 in ASM-deficient fibroblasts results in the ablation of ETV1 (*Figure 6A*), while ETV1 silencing has only a minor effect on KLF2 (*Figure 6A*). Since we have shown above that ETV1 is regulated at transcript level (*Figure 4—figure supplement 1A*), this result implies that KLF2 regulates (activates) the transcription of the gene encoding ETV1, in agreement with the increased transcript levels of ETV1 in ASM-deficient fibroblasts. These findings are also validated by the results of the KLF2 ChIP-Seq analysis, whose target

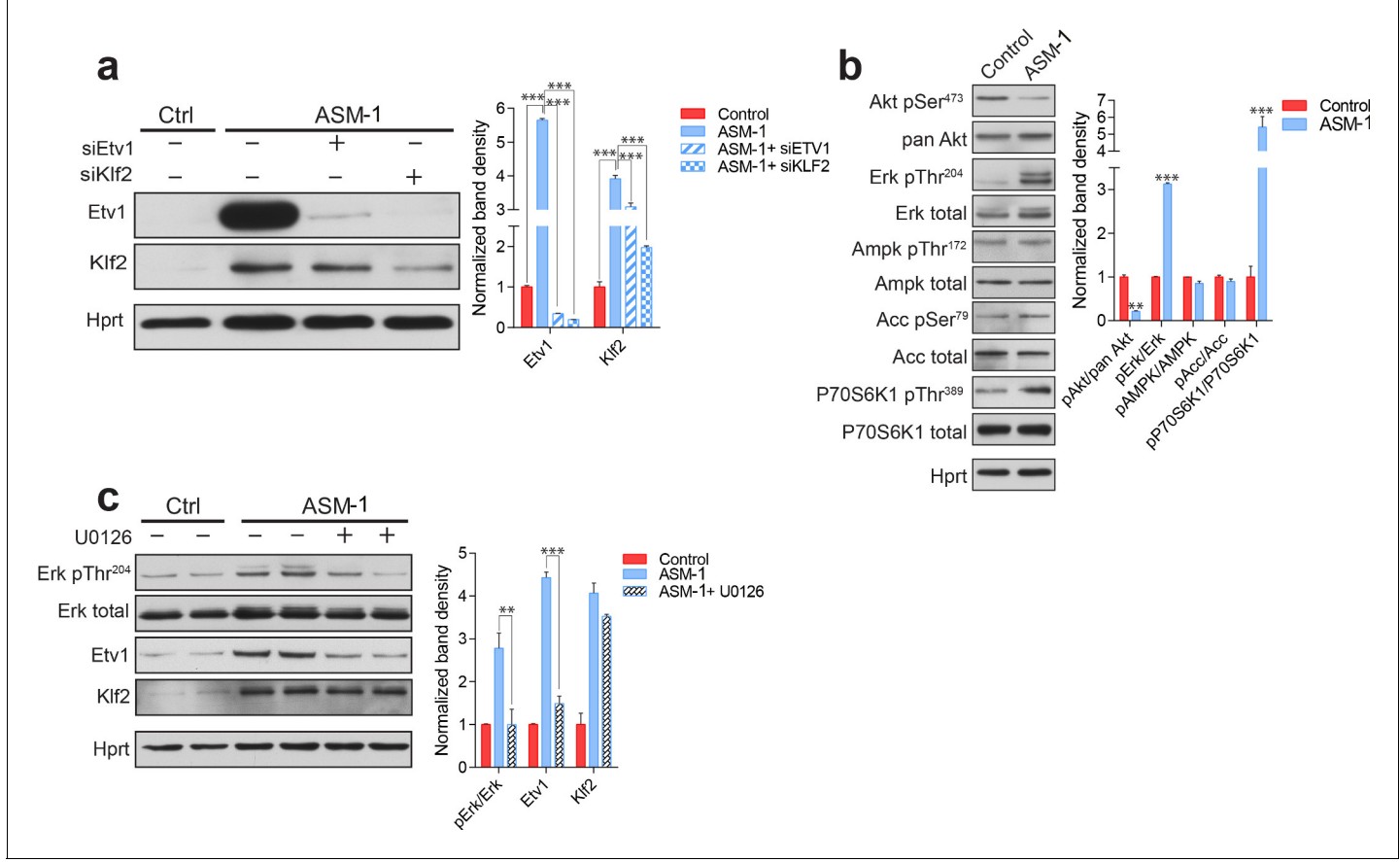

**Figure 6.** ETV1 up-regulation is dependent on KLF2 and ERK. (a) Silencing of KLF2 in ASM-deficient fibroblasts results in reduced levels of ETV1, shown by a representative western blot of whole cell extracts, with quantification of band densities (mean ± s.e.m, n = 3) in adjacent plots. One way ANOVA p-values ***p<0.001. (b) ASM-deficient fibroblasts show increased ERK and mTORC1 activities, reduced AKT activity and unchanged AMPK activity, as shown by a representative western blot of whole cell extracts with band density quantification presented in the adjacent plot as mean ± s.e.m., n = 3. T-test p-values **p<0.01 ***p<0.001 (c) ERK inhibition by treatment with U0126 (20 μM, 16 hr) in ASM-deficient fibroblasts results in reduced ETV1 levels but does not affect KLF2, as shown by a representative western blot, with band density quantification in the adjacent plot depicted as mean ± s.e.m. for biological triplicates. T-test p-values **p<0.01 ***p<0.001.
DOI: https://doi.org/10.7554/eLife.39598.021

genes include ETV1 (*Figure 4—figure supplement 4*). These results suggest that KLF2 and ETV1 are epistatic, with ETV1 downstream of KLF2.

Next, we tested known signaling modulators of KLF2 or ETV1 in ASM-deficient fibroblasts. Akt signaling down-regulates KLF2 (*Skon et al., 2013*), and we observed that Akt seems deactivated in ASM-deficient fibroblasts, as assessed by decreased phosphorylation of Akt Serine 473 (*Figure 6B*). ERK is a positive effector of ETV1 (*Janknecht, 1996*), and we found ERK signaling increased in ASM-deficient fibroblasts (*Figure 6B*). mTORC1 signaling is often involved in lysosomal stress signaling, and we found it activated in ASM-deficient fibroblasts, as assessed by the phosphorylation of p70S6 kinase (P70S6K) Threonine 389 (*Figure 6B*). AMPK signaling, which regulates mTORC1 as well as biogenesis of mitochondria and lysosomes, was not affected, as assessed by phosphorylation of AMPK target acetyl-CoA carboxylase (ACC) or of the activating phosphorylation of AMPK itself (*Figure 6B*). Inhibition of mTORC1 signaling in ASM-deficient fibroblasts by treatment with the mTORC1 inhibitor torin1 had no effect on the expression of mitochondria-related genes or mitochondrial function (data not shown).

We next tested if the increased ERK signaling was related to the increased levels of ETV1. We treated the ASM-deficient fibroblasts with the ERK inhibitor U0126, which led to the ablation of ERK signaling, as expected (*Figure 6C*). KLF2 was mostly unaffected by ERK inhibition (*Figure 6C*).

However, ETV1 was returned to control levels (*Figure 6C*). This result suggests that KLF2 can only trigger ETV1 expression in the presence of active ERK signaling.

## S1PR1 signaling dynamically regulates KLF2 and mitochondrial biogenesis and function

Next, we sought to identify the mechanism leading to KLF2 up-regulation. Since one of the consequences of lysosomal malfunction is the stalling of the autophagy pathway, we tested if KLF2 could be induced by perturbations in autophagy, such as inhibition of autophagosome formation (Atg5 silencing) or inhibition of the fusion of autophagosomes to lysosomes (syntaxin 17 silencing). However, no effect was observed in KLF2 (data not shown).

KLF2 is known to be negatively regulated by Akt signaling (*Skon et al., 2013*), which is repressed in ASM-deficient fibroblasts (*Figure 6B*). Interestingly, one of the genes induced by KLF2 is the sphingosine-1-phosphate receptor 1 (S1PR1) (*Skon et al., 2013*), which we find up-regulated at transcript level in ASM-deficient fibroblasts (*Figure 7—figure supplement 1*). S1PR1 and KLF2 are part of a signaling network in which the activity of the receptor represses its own expression by downregulating KLF2 via Akt activation (*Sinclair et al., 2008*; *Skon et al., 2013*). Interestingly, the S1PR1 receptor has been previously shown to affect mitochondrial function in T cells, but the mechanisms remained unexplored (*Mendoza et al., 2017*). Furthermore, the levels of sphingosine-1-phosphate (S1P) are decreased in the plasma of NPC1 patients (*Fan et al., 2013*), suggesting that signaling elicited by S1P may be down-regulated.

Given the connections between S1PR1, KLF2 and our findings implicating KLF2 in the regulation of mitochondrial-related gene expression, we decided to test if perturbation of the S1PR1 pathway in ASM- or NPC1-deficient cells could explain the up-regulation of KLF2 and, accordingly, the expression of mitochondria-related genes. To this end, we first sought to establish that S1PR1 can regulate mitochondrial biogenesis and function in healthy cells. We treated control fibroblasts with either a selective agonist (Sew2871) or with a selective inhibitor (W146) of S1PR1, and measured the effects on mitochondria. We observed that the activation of S1PR1 by the agonist Sew2871 results in increased transcript levels of mitochondria-related genes (*Figure 7A*). Reciprocally, inhibition of S1PR1 by W146 leads to decreased transcript levels of these genes (*Figure 7B*). Furthermore, activation of S1PR1 results in increased mitochondrial OCR under basal and uncoupled conditions (*Figure 7C*, quantified in 7E), while the inhibition of the receptor results in a robust inhibition of mitochondrial OCR (*Figure 7D*, quantified in 7F). Finally, we observed that KLF2 responds as expected to S1PR1 activity. When S1PR1 is activated, KLF2 levels decrease (*Figure 7G*), while inhibition of S1PR1 results in increased KLF2 abundance (*Figure 7H*). ETV1 shows a similar pattern, decreasing when S1PR1 is activated (*Figure 7G*) and increasing in response to S1PR1 inhibition (*Figure 7H*). Notably, the protein levels of mitochondrial proteins TFAM, cytochrome oxidase I (mtCOI),succinate dehydrogenase subunit b (SDHB) and porin (VDAC1) are all increased when KLF2 and ETV1 are down-regulated (S1PR1 activation, *Figure 7G*), and all decreased when KLF2 and ETV1 levels are increased (S1PR1 inhibition, *Figure 7H*). These results underscore that the S1PR1-KLF2-ETV1 mitochondrial biogenesis pathway can be dynamically regulated in control fibroblasts. Furthermore, these data suggest that the S1PR1 pathway may be down-regulated in ASM-deficient fibroblasts, given the increased levels of KLF2 and the decreased expression of mitochondria-related genes. Interestingly, the expression of sphingosine kinase 1 (*SPHK1*), which generates S1P that can be exported to the extracellular space, is down-regulated in ASM-deficient fibroblasts (*Figure 7—figure supplement 1*). Similarly, *SPHK2*, which generates S1P intracellularly, in mitochondria and endoplasmic reticulum, is also down-regulated in ASM-deficient fibroblasts (*Figure 7—figure supplement 1*). Altogether, these results suggest that S1P signaling via S1PR1 is profoundly down-regulated in ASM-deficient fibroblasts, and that this event is at the root of the up-regulation of KLF2 and its downstream consequences, particularly ETV1 induction and inhibition of mitochondrial biogenesis.

## S1PR1 is mislocalized in ASM-deficient cells and unresponsive to activators

Given the apparent down-regulation of S1PR1 signaling in ASM deficiency, we set to test if reactivation of the S1PR1 pathway in ASM-deficient fibroblasts would rescue the expression of

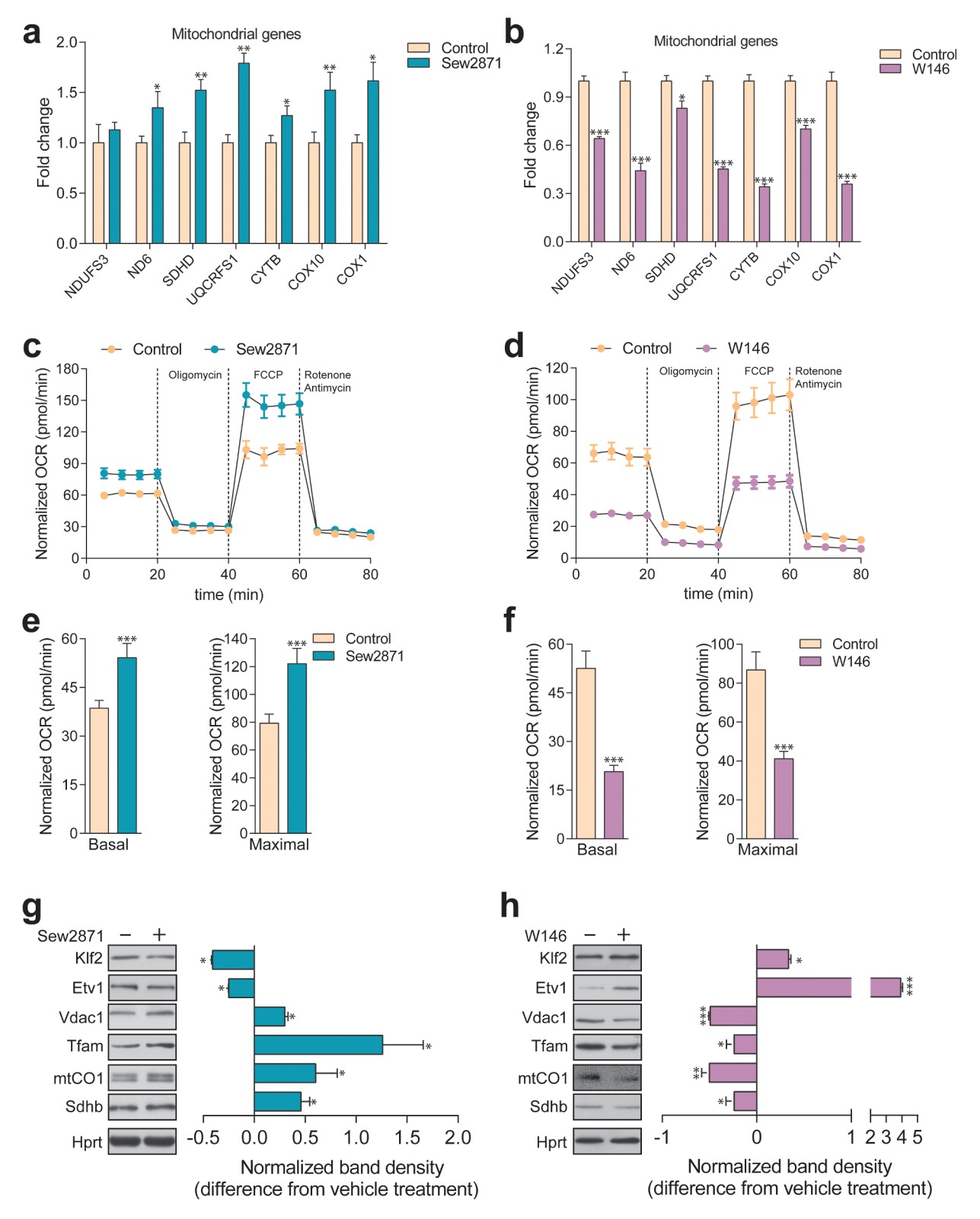

**Figure 7.** Dynamic regulation of S1PR1 activity impacts mitochondrial biogenesis and function. (**a**) Transcript levels of mitochondrial-related genes increase upon activation of S1PR1 with the agonist Sew2871 (5 µM, 16 hr; DMSO as vehicle control), as measured by qPCR. Plots show mean ± s.e.m., n = 3. T-test p-values *p<0.05 and **p<0.01 (**b**) Transcript levels of mitochondrial-related genes decrease upon inhibition of S1PR1 with the competitive antagonist W146 (10 µM, 16 hr; methanol as vehicle control), as measured by qPCR. Plots show mean ± s.e.m., n = 3. T-test p-values **p<0.01 and

*Figure 7 continued on next page*

*Figure 7 continued*

***p<0.001 (c) Increased OCR in cells treated with the S1PR1 agonist Sew2871 compared to vehicle control (DMSO), quantified in panel (e). (d) Decreased OCR in cells treated with the S1PR1 antagonist W146 compared to vehicle control (methanol), quantified in panel (f). Quantifications in e and f represent mean ± s.em., n = 3 with T-test p-values ***p<0.001. (g) Representative blots showing decreased protein levels of KLF2 and ETV1, and increased amounts of mitochondrial proteins VDAC1, TFAM, CO1 and SDHB, in cells treated with S1PR1 agonist Sew2871, assessed by western blots of whole cell extracts, using HPRT as loading control. Adjacent plot depicts the fold difference in band density relative to vehicle control (DMSO) as mean ± s.e.m., n = 2 with technical triplicates (the line on zero denotes no change relative to the controls, negative numbers show decrease in fold change, positive numbers show increased fold change). T-test p-value *p<0.05 (h) Representative blots depicting increased protein levels of KLF2 and ETV1, and decreased amounts of mitochondrial proteins VDAC1, TFAM, CO1 and SDHB, in cells treated with S1PR1 antagonist W146, assessed by western blots of whole cell extracts, using HPRT as loading control. Adjacent plot shows the difference in fold band density compared to vehicle control (methanol) and depicted as average ± s.e.m., n = 2 with technical triplicates. T-test p-value *p<0.05, **p<0.01 and ***p<0.001.
DOI: https://doi.org/10.7554/eLife.39598.022

The following figure supplement is available for figure 7:

**Figure supplement 1.** Transcript levels of sphingosine-1-phosphate receptor 1 (S1PR1) and sphingosine kinases 1 (SPHK1) and 2 (SPHK2) in ASM-deficient fibroblasts.
DOI: https://doi.org/10.7554/eLife.39598.023

mitochondria-related genes as well as mitochondrial function. We treated control and ASM-deficient fibroblasts with the S1PR1 agonist Sew2871, and in agreement with our data shown above (*Figure 7A*), we found an increase in the expression of mitochondria-related genes in control fibroblasts (*Figure 8B*). However, and surprisingly, the ASM-deficient fibroblasts did not respond to the treatment with the S1PR1 agonist: no change was observed in the transcript levels of mitochondria-related genes (*Figure 8B*). Similar results were obtained when using S1P instead of the agonist (data not shown). These results suggest that the S1PR1 receptor is absent or inaccessible to extracellular cues, implying that it may be sequestered away from the plasma membrane. The protein levels of S1PR1 are not changed in ASM-deficient fibroblasts (*Figure 8C*). Therefore, we tested if S1PR1 localization at the plasma membrane was affected in ASM-deficient cells. We used a PE-conjugated antibody against S1PR1 for flow cytometry, in non-permeabilized cells, and determined the amount of plasma membrane labelling in control and ASM-deficient fibroblasts. As negative control, we treated cells with FTY720, which antagonizes S1PR1 signaling by promoting its endocytosis. The treatment with FTY720 reduced the levels of S1PR1 at the plasma membrane, which were robustly decreased in ASM-deficient cells. Thus, the mislocalization of S1PR1 in ASM-deficient cells, and consequent decreased signaling, explain the increase in KLF2 signaling and its downstream consequences.

## Upregulation of KLF2 and ETV1 has a protective effect in ASM-deficient cells

Finally, we sought to test if induction of KLF2 and ETV1 in ASM-deficient cells contributes to Niemann Pick disease pathology or if it is a protective mechanism. We again resorted to the silencing of KLF2 and ETV1 by siRNA, and measured cell death using the Annexin-V/propidium iodide flow cytometry assay. We observed that ~38% of ASM-deficient cells are apoptotic (high Annexin V signal), while this number increases to ~55% when KLF2 or ETV1 are silenced (*Figure 8E*). Accordingly, the amount of cleaved (active) caspase-3 and of cleaved PARP (caspase-3 target) are increased when KLF2 and ETV1 are silenced (*Figure 8F*). Finally, cell viability, measured by the CellTiter-Glo assay, is robustly decreased when KLF2 or ETV1 are silenced (*Figure 8G*). Thus, the up-regulation of KLF2 and ETV1 in ASM-deficient cells is a protective mechanism.

## Discussion

This study addresses a novel mechanism by which mitochondria are impaired in lysosomal lipid storage diseases. We show here that the transcription factors KLF2 and ETV1 repress the expression of genes encoding mitochondrial proteins. Both KLF2 and ETV1 are up-regulated in patient cells from Niemann-Pick type C and acid sphingomyelinase (ASM) deficiency, and their silencing, particularly KLF2, is sufficient to return mitochondrial biogenesis and function to control levels. Decreased signaling through sphingosine-1-phosphate receptor 1 (S1PR1) activates KLF2, which induces the expression of ETV1, culminating in the down-regulation of mitochondrial biogenesis.

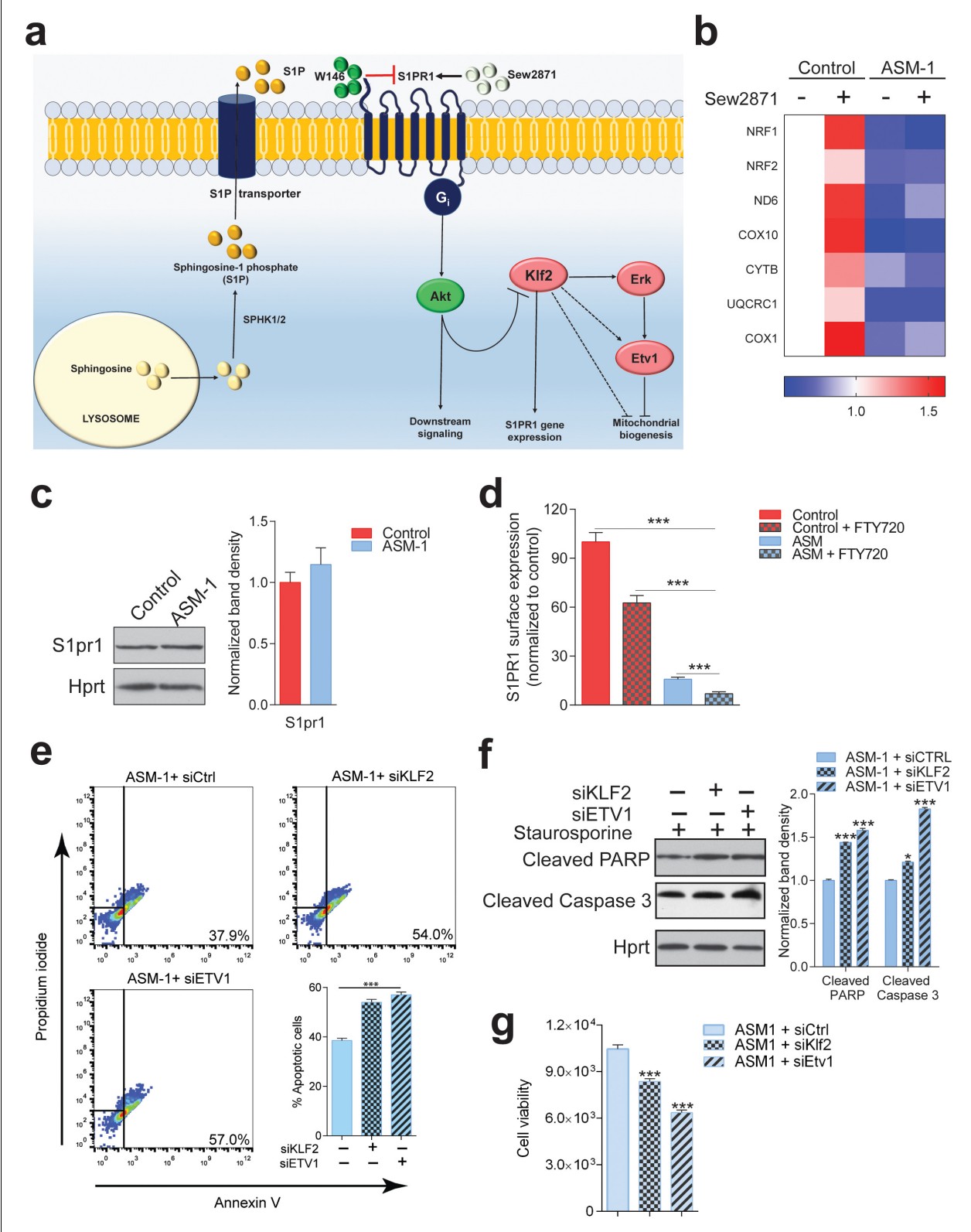

**Figure 8.** S1PR1 signaling in Niemann-Pick disease. (**a**) Schematic illustration of sphingosine-1-phospate (S1P) signaling. S1P is generated from sphingosine by the kinases SPHK1 (plasma membrane) and SPHK2 (endoplasmic reticulum and mitochondria), and can be transported out of the cell. Extracellular S1P can activate several receptors (S1PR1-5). Specifically, stimulation of S1PR1 triggers Akt signaling which regulates KLF2 levels. Expression of S1PR1 is regulated by KLF2, which as shown by our data also activates ETV1. Sew2871 is an agonist of S1PR1, and W146 is an antagonist.

*Figure 8 continued on next page*

*Figure 8 continued*

(**b**) Treatment of control fibroblasts with S1PR1 agonist Sew2871 (5 μM, 16 hr) results in increased transcript levels of mitochondria-related genes in control fibroblasts, but has no effect on ASM-deficient fibroblasts. The data is presented in a Heatmap for n = 3, in which blue denotes decrease in expression compared to the control cells (white represents no change relative to the control values) and red denotes increase. Note the mostly increased (red) mitochondrial genes in control fibroblasts treated with Sew2871, while blue in ASM fibroblasts regardless of the treatment. (**c**) Protein levels of S1PR1 are not changed in ASM fibroblasts, as measured by western blot using whole cell extracts. Adjacent plot shows the quantification presented as mean ± s.e.m., n = 3. T-test p-value>0.05. (**d**) Staining of S1PR1 present at the plasma membrane, in non-permeabilized cells, measured by flow cytometry. FTY720 triggers the endocytosis if S1PR1 and was used as a negative control for surface staining. Note the barely detectable surface S1PR1 levels. Plots represent the average fraction of S1PR1 levels normalized to vehicle treated control cells and depicted as mean ± s.e.m., n = 3. T. test p-value ***p<0.001 (**e**) Staurosporine-treated ASM-deficient cells with KLF2 and ETV1 silencing show increased apoptotic cell population relative to control ASM-deficient cells as measured by flow cytometry with Annexin-V and Propidium iodide staining. Quantifications are depicted as mean ± s.e. m., n = 5. T-test p-value ***p<0.001. (**f**) Staurosporine-treated ASM-deficient fibroblasts with either silencing control, KLF2 or ETV1 knockdowns show increased protein amounts of Cleaved PARP and Cleaved Caspase three levels in cells with KLF2 or ETV1 silencing and quantifications in adjacent graphs show mean ± s.e.m., n = 3. T-test p-value **p<0.01 and ***p<0.001. (**g**) Decreased cell viability as measured by Glo Titer Assay in ASM-deficient fibroblast with either KLF2 or ETV1 silencing. Plots represent mean± s.e.m., n = 2 with six technical replicates per condition. T-test p-value, ***p<0.001.

DOI: https://doi.org/10.7554/eLife.39598.024

The transcriptional regulation of mitochondrial biogenesis is known since the identification of the transcription factor nuclear respiratory factor 1 (NRF1), which induces the expression of many respiratory chain and mtDNA maintenance genes (*Scarpulla et al., 2012*). Several other transcription factors have been shown to stimulate mitochondrial biogenesis, such as estrogen related receptor α (ERRα) or the oncogene myc (*Scarpulla et al., 2012*). The role of the co-activator PGC1α (peroxisome proliferation activated receptor gamma, co-activator 1α) has also been shown to promote NRF1- and ERRα-mediated mitochondrial biogenesis (*Wu et al., 1999*). However, to our knowledge, no transcription factor has previously been shown to repress mitochondrial biogenesis. Thus, the roles of KLF2 and ETV1 as repressors of mitochondrial biogenesis, shown in this manuscript, open a new paradigm on the transcriptional regulation of the mitochondrial biogenesis. Interestingly, another Krüppel-like factor, KLF4, was recently shown to promote mitochondrial biogenesis in the heart (*Liao et al., 2015*), implying that the repressive behavior of KLF2 is a specificity of this transcription factor and not a characteristic transversal to the whole Krüppel-like factor family.

Notably, the transcription factor NRF1, which is a known positive regulator of mitochondrial biogenesis and is down-regulated in fibroblasts with acid sphingomyelinase deficiency, is also repressed by KLF2 and ETV1. It therefore seems that KLF2, ETV1 and NRF1 may form a transcriptional regulatory network that dynamically regulates mitochondrial biogenesis, with 'accelerator' (NRF1) and 'brakes' (KLF2 and ETV1). The transcriptional network between KLF2, ETV1 and NRF1, as well as the involvement of other transcription factors such as ERRα, myc, or co-activators such as PGC1α, warrants further research.

It is particularly interesting that a transcriptional network repressing mitochondrial biogenesis appears robustly active in lysosomal diseases. The role of lysosomes in cellular function has been subject of increasing attention, both regarding its physiological roles as a signaling platform as well as the pathological consequences of lysosomal defects in lysosomal storage diseases (*Settembre et al., 2008*; *Ballabio and Gieselmann, 2009*; *Perera and Zoncu, 2016*; *Platt et al., 2012*). Numerous studies describe the impact of lysosomal defects on the function of other organelles, particularly mitochondria, in several lysosomal storage diseases (*Diogo et al., 2018*; *Plotegher and Duchen, 2017*; *Raimundo et al., 2016*; *Torres et al., 2017*). Mitochondria are usually impaired in cells and tissues with primary lysosomal defects, with decreased oxygen consumption and increased production of superoxide and other reactive oxygen species (*Jolly et al., 2002*; *Plotegher and Duchen, 2017*). However, this is often usually attributed to a decrease in autophagy (and mitophagy), with the consequent accumulation of damaged mitochondria in the cytoplasm. Our data in cellular and mouse models of Niemann-Pick-C disease and acid sphingomyelinase deficiency shows, however, that in addition to defective autophagy there is a signaling mechanism based on the induction of two transcription factors, KLF2 and ETV1, which repress mitochondrial biogenesis. This may represent a signaling circuit in which the cells with lysosomal defects repress the generation of an organelle whose degradation requires lysosomal function. It may also be a consequence of the accumulation of lipids such as sphingomyelin and cholesterol in the lysosomes in Niemann-Pick type

C and acid sphingomyelinase deficiency, which is likely to result in deficiency of those lipids in other cellular locations. Thus, one conceivable cellular adaptation would be shutting down the mitochondrial respiratory chain and citrate cycle, which would allow to shunt citrate to the cytoplasm, where it can be converted by acetyl-CoA lyase to acetyl-CoA and used for lipid synthesis (*Bauer et al., 2005*; *Wellen et al., 2009*). Interestingly, a reciprocal mechanism seems to exist, since chronic mitochondrial defects result in repression of lysosomal biogenesis (*Nezich et al., 2015*; *Woś et al., 2016*; *Fernández-Mosquera et al., 2017*) and function (*Demers-Lamarche et al., 2016*; *Fernandez-Mosquera et al., 2018*). The interplay between mitochondria and lysosomes is a relatively novel concept that is only now being grasped (*Diogo et al., 2018*; *Raimundo et al., 2016*). The existence of cross-talk mechanisms involving transcriptional networks implies that the communication between these two organelles goes beyond metabolic cues, and involves complex cellular signaling.

The up-regulation of KLF2 in ASM-deficient cells seems to be a consequence of impaired sphingosine-1-phosphate (S1P) signaling through S1P receptor 1 (S1PR1). This receptor had previously been implicated in the regulation of mitochondrial function in T cells, but the mechanism remained unclear (*Mendoza et al., 2017*). We show in this study that S1PR1 is a bona fide bi-directional regulator of mitochondrial function via the effect of KLF2 and ETV1 on mitochondrial biogenesis. Indeed, both the activation and the inhibition of S1PR1 in control cells impacted mitochondrial biogenesis and function. This effect can be interpreted in diverse biological scenarios. For example, in the context of the role of S1P and S1PR1 in angiogenesis, decreased signaling could be interpreted as impaired angiogenesis, thus inefficient delivery of $O_2$, and the cells respond by shutting down the major $O_2$ consumption cellular component – mitochondria. Interestingly, the acid sphingomyelinase-deficient fibroblasts were non-responsive to agonists of S1PR1, which suggests that the receptor may be sequestered away from the plasma membrane in the patient cells. In support of this hypothesis, the amount of S1PR1 in the plasma membrane of the ASM-deficient cells is negligible, while the total protein levels of S1PR1 are similar to control cells. This result implies a mistargeting of S1PR1 in acid sphingomyelinase deficiency and Niemann-Pick disease, which is akin to other proteins aberrantly mislocalized away from the plasma membrane in these diseases, such as Met receptor tyrosine kinase or K-Ras (*Schuchman and Wasserstein, 2016*; *Praggastis et al., 2015*). Thus, a therapeutic strategy targeting the receptor activity would likely be insufficient.

The contribution of the signaling pathways mediating communication between mitochondria and lysosomes and their roles in pathology certainly warrants further exploration, not just in mitochondrial and lysosomal diseases but also in the context of neurodegenerative diseases that arise from defects of either of these organelles.

# Materials and methods

## Drugs and cellular treatments

The following drugs were used for cellular treatments: 1 µM Oligomycin (Sigma, O4876), 2 µM Carbonyl cyanide 3-fluorophenylhydrazone (FCCP) (Sigma, C2920), 1 µM Rotenone (Sigma, R8875), 1 µM Antimycin (Sigma, A8674), 40 µM Desipramine (Biotrend, BG0162), 5 µM Sew2871 (Cayman, 10006440), 20 µM U0126 (Millipore, 662005), 10 µM U18666A (Cayman, 10009085), 10 µM W146 (Sigma-Aldrich, W1020), 2 µM FTY720 (Selleckchem, S5002) and 4 µM Staurosporine (Sigma-Aldrich, 37095).

## Cell culture and transient transfections

Control and Niemann-Pick patient fibroblasts were grown in DMEM high glucose medium (Gibco, 11965) supplemented with 10% fetal bovine serum and 1% Penicillin/Streptomycin at 37°C and 5% CO2, in a humidified incubator, unless otherwise stated. ASM-1 patient fibroblasts retained about 5% of the control activity of acid sphingomyelinase, and were collected and maintained according to the ethical guidelines of the UMG. Control and patient fibroblasts were transfected with siRNAs for ETV1 or KLF2 using electroporation (Amaxa kit, Lonza, V4XP-1024) or with scrambled control siRNA following manufacturer's protocol. Additional control, human, adult primary fibroblasts were obtained ATCC (PCS-201–012). NPC1 patient cells and an additional ASM patient line (referred to as ASM-2 in the text) were obtained from Coriell Institute for Medical Research (GM18398, GM13205). The cell lines were not authenticated for cross-type contamination and were tested

periodically for mycoplasma. The use of human cells for these studies was approved by the Ethical Commission of the Universitätsmedizin Göttingen.

## Mouse tissues

The NPC1 mice are maintained at the University of Helsinki, and the ASM mice are maintained at the University of Erlangen. In both cases, the maintenance of these animals is approved under the directive 2010/63/EU.

## XF medium

XF assay medium (Seahorse Bioscience, 100965–000) was supplemented with sodium pyruvate, glutamax and glucose following manufacturer's recipe and the pH of medium was adjusted to 7.4.

## Oxygen consumption rate measurements

OCR was measured in fibroblasts using the XF96 Extracellular Flux analyzer (Seahorse Bioscience). Briefly, cells were seeded at $2 \times 10^4$ cells per well in XF96 cell culture multi-well plates in DMEM medium and incubated for 24 hr in the growth conditions stated for all cell cultures. XF96 cartridges were incubated overnight in XF calibrant at 37°C in a non-CO2 incubator. Prior to OCR measurements, the growth medium of cells was exchanged with XF medium and incubated at 37°C in a non-CO2 incubator for 1 hr. Inhibitors were diluted to appropriate concentrations in XF medium and loaded into corresponding microwells in the XF96cartridge plate. Following equilibration of sensor cartridges, XF96 cell culture plate was loaded into the XF96 Extracellular Flux analyzer at 37°C and OCR was measured after cycles of mixing and acquiring data (basal) or inhibitor injection, mixing and data acquisition.

## Western blotting

Whole cell extracts of cultured fibroblast were prepared in 1.5% n-dodesylmaltoside (Roth, CN26.2) in PBS supplemented with protease and phosphatase inhibitor cocktail (Thermoscientific, 78442) as described (*Raimundo et al., 2009*). Protein concentrations of whole cell extracts were determined using a Bradford assay (Bio-Rad, 500–0006). 50 µg of sample proteins per well were subjected to Sodium dodecyl sulfate -polyacrylamide gel electrophoresis (SDS-PAGE) and transferred to polyvinylidene fluoride (PVDF) membranes (Amersham, Life Technologies). After blocking in 5% Milk in TBS tween, membranes were immunoblotted with the following antibodies: Sqstm1 (Abcam, ab110252), Hprt (Abcam, ab10479), Klf2 (Abcam, ab203591), Etv1 (Abcam, ab184120), Lc3b (Cell signaling, 3868), Pan Akt (Cell signaling, 4691), Phospho Akt (Cell signaling, 4060), Total Erk1/2(Cell signaling, 4695), Phospho Erk1/2 (Cell signaling, 4376), Tfam (Abcam, ab138351), P70s6k1 (Cell signaling, 2708), Phospho P70s6k1 (9234), Acc (Cell signaling, 3676), Phospho Acc (Cell signaling, 3661), Ampkα (Cell signaling, 5832), Phospho Ampkα (Cell signaling, 2535), Nrf1 (Abcam, ab175932), Atp5a, Uqcrc2, Sdhb and mtCO1 OXPHOS cocktail (Abcam, ab110413), S1pr1 (Abcam, ab125074), Cleaved PARP (Cell signaling, 5625), Cleaved Caspase-3 (Cell signaling, 9664), Laminin A/C (Cell signaling,) and Vdac1 (Abcam, ab14734). Band densitometric quantifications were determined using ImageJ software 1.48 v. Following normalization with Hprt, all control samples of each experiment were centered at one to ease relative comparisons with experimental samples.

## Subcellular fractionation

Patient and control fibroblasts were harvested at 80% confluence by scraping in ice-cold PBS. Nuclear and cytosolic fractions were isolated from the cell pellets using a nuclei/cytosol fractionation kit (BioVision, K266). Nuclear and cytosolic proteins, along with whole cell extracts, were subjected to Western blot analyses.

## Measurement of lysosomal proteolytic capacity

Lysosomal proteolytic capacity was measured using the DQ Red BSA Dye (Molecular Probes, D-12051) following manufacturer's protocol. Briefly, 100 ul of 1mg/ml dye was added to 10 ml of warm DMEM medium. Previously plated cells in a transparent 96 well-plate were loaded with 100 ul per well each of the dye containing medium and incubated at 37°C for 1 hr. Cells were then washed twice with warm PBS and the medium was replaced with 100 µL/well of warm EBSS medium. The

kinetics of DQ Red BSA digestion were recorded at respective excitation and emission maxima of 590 nm and 620 nm in a multi-plate reader over a 4 hr period.

## Quantitative RT-PCR

RNA extraction and purification from fibroblasts were performed using Crystal RNA mini Kit (Biolab, 31-01-404). From mouse livers, RNA was extracted using the TRI Reagent (Sigma-Aldrich, T9424). RNA concentration and quality were determined using Nanodrop (PeqLab) and cDNA was synthesized with iScript cDNA synthesis kit (Bio-Rad, 178–8991) following manufacturer's protocol. Each 8 µl q-PCR was made of 4 µl diluted cDNA, 0.2 µl of each primer (from 25 µM stock) and 3.6 µl of iTaq Universal Sybr Green Supermix (Bio-Rad, 172–5124) and ran on the QuantStudio 6 Flex Real-Time PCR system (Applied Biosystems). Transcript levels measured by quantitative PCR (qPCR) were determined by the ΔΔCT method using HPRT and GAPDH (not shown) as reference genes. Unless otherwise indicated, qPCR experiments of at least three biologically independent experiments always included at least technical triplicates. For the determination of relative expression (fold change), all control samples were centered at one by normalizing the expression of experimental samples to those of the corresponding controls.

## Flow cytometry

Measurement of mitochondrial superoxide levels using MitoSOX Red Mitochondrial superoxide indicator (Molecular Probes, M36008) was performed by flow cytometry according to the manufacturer's instructions. For S1PR1 plasma membrane localization, $1 \times 10^6$ control and ASM deficient fibroblasts treated with or without 2 µM FTY720 were labelled in suspension with 10 µL of PE-conjugated S1PR1 antibody (R and D systems, FAB2016P) for 1 hr, washed twice in isotonic PBS supplemented with 1% BSA, resuspended in 200–400 uL of buffer and subjected to flow cytometry analyses for the surface expression of S1PR1. For apoptosis measurements, $1 \times 10^5$ cells were plated 24 hr prior to flow cytometric determinations. Cells were then treated for 1 hr with 4 µM Staurosporine, harvested and stained in suspension with Annexin V (BD Pharmingen, 556419) and Propidium iodide (Sigma-Aldrich, P4170) in the dark for 20 min and analyzed by flow cytometry. Analyses of flow cytometry results were done using FlowJo v10 (FlowJo, LLC).

## Cell viability

Measurement of cell viability in patient fibroblast was carried out using the Cell Titer-Glo Luminiscent cell viability assay (Promega, G7570) following manufacturer's protocol.

## Dataset selection

In order to identify transcriptional signatures mediating interactions between organelles in Niemann-Pick pathology, we mined for microarray data involving Niemann-Pick mouse models from the Gene Expression Omnibus (http://www.ncbi.nlm.nih.gov/geo). Criteria for dataset selection included datasets with multiple replicates from several tissues. The dataset selected was GSE39621, which includes samples of brain, liver and spleen of mice before and after 6 weeks of age, when the symptoms of the disease start manifesting. Given that the spleen may contain immune cells in addition to splenocytes, and likely to have many more of non-splenocytes in the disease case, since spleen enlargement is a hallmark of the disease, we considered that the control and $Npc1^{-/-}$ were not directly comparable and thus used only the data relative to brain and liver.

## Organelle-specific gene lists

We obtained organelle proteomes from up-to-date and comprehensive databases for mitochondrial (and respiratory chain subunits), lysosomal, peroxisomal, endoplasmic reticulum and Golgi proteomes (*Table 1*). These protein IDs were converted to NCBI gene symbols, which were then used to identify the corresponding probeset names for different microarray matrices.

## Microarray data analysis

We obtained mouse *Npc1* wildtype, $Npc1^{+/-}$ and $Npc1^{-/-}$ in asymptomatic (less than 6 weeks old) and symptomatic (more than 6 weeks old) brain, liver and spleen from the GEO database (*Alam et al., 2012*). The controls for the NPC1 dataset are the wt mice in the brain but the

heterozygous mice in the other tissues. We used the software GeneSpring (Agilent Technologies, Santa Clara, CA) to normalize the datasets by robust multi-array averaging (RMA) to normalize datasets (*Raimundo et al., 2009*). The datasets for all tissues originating from the same knock-out mouse and corresponding controls were normalized together. After normalization, we determined which transcripts had significantly different expression between *Npc1*$^{-/-}$ and controls for each individual tissue, using ANOVA. We also calculated the fold change from probe expression values between lysosomal disease and control mice for each tissue. The statistical filter was set at p-value<0.05, and the transcripts that pass the filter for each tissue represent the corresponding transcriptional signature.

To calculate the average expression of organelle-specific gene lists, we normalized each transcript to the average of the control samples, and calculated the average of the expression levels of all genes in each organelle-specific gene list. To determine if the difference observed between *Npc1*$^{-/-}$ and controls was significant, we calculated the t-test p-value (unpaired, unequal variance) for the whole gene set using Microsoft Excel. Given that the lists have hundreds of genes, we performed a Bonferroni post-hoc correction. The adjusted p-values<0,05 were considered significant.

### Pathway analysis and identification of transcriptional regulators

We employed a multi-dimensional strategy aimed at the identification of signaling pathways, as described (*Raimundo et al., 2012*; *Raimundo et al., 2009*; *Schroeder et al., 2013*; *West et al., 2015*). The transcriptional lists were imported to the software Ingenuity Pathway Analysis (IPA) (http://www.ingenuity.com), which then determines which pathways and transcriptional regulators are statistically enriched, using Fisher's exact test. The statistical threshold was set at p<0.01.

### Promoter analysis

To perform promoter analysis on the respiratory chain genes, we imported the respiratory chain gene list to the software Genomatix Suite (www.genomatix.de). Then we set a pipeline within the software suite, by first defining the promoters of the respiratory chain genes and then determining which transcription factors (TF) had binding sites on them. To locate the promoters, we use the Genomatix tool Gene2Promoter, and defined the promoter region from 500 base pairs upstream (−500) the transcription start site (TSS) until 100 base pairs downstream the TSS (+100). Given that some genes may have more than one promoter due to alternative splicing, we selected only the promoters that drive the expression of the transcript leading to the protein that functions as a respiratory chain subunit. The promoter sequences were then used to determine cis-elements and identify the corresponding TF, limiting the search to those TF that had at least a binding site in at least 85% of the promoters. The software provides a statistical assessment of the enrichment of the binding sites for each TF family in the promoters under analysis. We set a threshold of p<0.05 for the Fisher's exact test p-value for each TF family enrichment. Then, we determine, for each significantly enriched family, which individual TF are included, and select as relevant TF those that have a binding site in at least 50% of the promoters under analysis.

### Statistical analysis

Statistical analyses were carried out using Graph Pad Prism 6 and 7 softwares. Unless otherwise stated in the corresponding figure legends, all measures throughout this manuscript were summarized as graphs displaying mean ± s.e.m., of at least three independent biological replicates. The means of the corresponding controls are typically centered at one to ensure easier comparisons unless otherwise stated. For in vivo experiments, n represents number of mice of each genotype used in this study. For cell culture work, n refers to the number of independent experiments carried out with different stocks of each cell line. Each n included at least technical duplicates for cells and the standard errors of the means were calculated from the means of the numbers of independent biological replicates (n) with their technical replicates. Differences between group means were determined by the unpaired Welch's t-test, assuming unequal variances between two groups and One way ANOVA for multi-group (at least three) comparisons; *p<0.05; **p<0.01; ***p<*0.001*; ns, nonsignificant p>*0.05*.

## Accession numbers

The publicly-available transcriptome datasets used in this study are GSE39621 for Niemann Pick's disease mouse model (*Npc1*[-/-]) (*Alam et al., 2012*) and GSE27602 for *Klf2*[-/-] mice (*Redmond et al., 2011*). The accession numbers for the ETV1 ChIP-ChIP (*Baena et al., 2013*) and KLF2 ChIP-Seq (*Yeo et al., 2014*) datasets are GSE39388 and E-MTAB-2365 respectively.

## Acknowledgements

This research was supported by ERC Starting Grant 337327 and AMDA Research Grant (NR); Deutsche Forschungsgemeinschaft Emmy-Noether Award and Schram Stiftung Grant (IM); Deutsche Forschungsgemeinschaft SFB1190 (NR and IM); Academy of Finland Grant 312491 (EI); Deutsche Forschungsgemeinschaft (GRK2162/1), Neurodevelopment and Vulnerability of the Central Nervous System (CM).

We thank Dr. Ralf Janknecht for the ETV1 constructs and Dr. Roberto Zoncu for the NPC1 constructs.

## Additional information

### Funding

| Funder | Grant reference number | Author |
| --- | --- | --- |
| Deutsche Forschungsgemeinschaft | GRK2162/1 | Christiane Mühle |
| Academy of Finland | 312491 | Elina Ikonen |
| Deutsche Forschungsgemeinschaft | SFB1190 | Ira Milosevic Nuno Raimundo |
| Deutsche Forschungsgemeinschaft | Emmy-Noether Award | Ira Milosevic |
| Schram Stiftung | | Ira Milosevic |
| H2020 European Research Council | 337327 | Nuno Raimundo |
| Acid Maltase Deficiency Association | Research Grant | Nuno Raimundo |

The funders had no role in study design, data collection and interpretation, or the decision to submit the work for publication.

### Author contributions

King Faisal Yambire, Data curation, Formal analysis, Validation, Investigation, Methodology, Writing—original draft, Project administration, Writing—review and editing; Lorena Fernandez-Mosquera, Investigation, Methodology; Robert Steinfeld, Christiane Mühle, Resources, Validation; Elina Ikonen, Resources, Formal analysis, Methodology; Ira Milosevic, Validation, Methodology; Nuno Raimundo, Conceptualization, Resources, Data curation, Formal analysis, Supervision, Funding acquisition, Validation, Investigation, Methodology, Writing—original draft, Project administration, Writing—review and editing

### Author ORCIDs

King Faisal Yambire https://orcid.org/0000-0003-2417-450X
Lorena Fernandez-Mosquera http://orcid.org/0000-0003-4606-5056
Christiane Mühle https://orcid.org/0000-0001-7517-9154
Ira Milosevic http://orcid.org/0000-0001-6440-3763
Nuno Raimundo http://orcid.org/0000-0002-5988-9129

## Ethics

Animal experimentation: All mice were handled according to the rules of law under application at the University of Helsinki (Finland) and at the University of Erlangen (Germany), according to the directive 2010/63/EU.

## Decision letter and Author response

Decision letter https://doi.org/10.7554/eLife.39598.040
Author response https://doi.org/10.7554/eLife.39598.041

# Additional files

## Supplementary files

• Supplementary file 1. Results of the promoter analysis of lysosomal genes. The transcription factor families that passed the significance threshold (Fisher exact test p<0.01) and the respective p-values are indicated.
DOI: https://doi.org/10.7554/eLife.39598.025

• Supplementary file 2. Mitochondrial KLF2 target genes obtained from the analysis of dataset E-MTAB-2365. These targets represent the common genes between the complete KLF2 target list and the mitochondrial gene list.
DOI: https://doi.org/10.7554/eLife.39598.027

• Transparent reporting form
DOI: https://doi.org/10.7554/eLife.39598.030

• Supplementary file 3. List of the transcription factors predicted to be significantly activated in the liver and brain of NPC1 KO mice compared to WT. The transcription factors and respective p-value is indicated. Transcription factors labelled in red were found to be significantly involved in both liver and brain of NPC1 KO, and thus selected for further analysis.
DOI: https://doi.org/10.7554/eLife.39598.026

• Supplementary file 4. qPCR primers.
DOI: https://doi.org/10.7554/eLife.39598.028

• Supplementary file 5. siRNA sequences.
DOI: https://doi.org/10.7554/eLife.39598.029

## Data availability

The publicly-available transcriptome datasets used in this study are GSE39621 for Niemann-Pick's disease mouse model ($Npc1^{-/-}$) (Alam et al., 2012) and GSE27602 for $Klf2^{-/-}$ mice (Redmond et al., 2011). The accession numbers for the ETV1 ChIP-ChIP (Baena et al., 2013) and KLF2 ChIP-Seq (Yeo et al., 2014) datasets are GSE39388 and E-MTAB-2365 respectively.

The following previously published datasets were used:

| Author(s) | Year | Dataset title | Dataset URL | Database and Identifier |
|---|---|---|---|---|
| Alam S, Shin J, Tamez P, Haldar K | 2012 | Expression data from brain, liver and spleen of Npc1-/- mice | https://www.ncbi.nlm.nih.gov/geo/query/acc.cgi?acc=GSE39621 | NCBI Gene Expression Omnibus, GSE39621 |
| Redmond LC, Dumur CI, Archer KJ, Grayson DR, Haar JL, Lloyd JA | 2011 | KLF2-Regulated Gene Expression in Mouse Embryonic Yolk Sac Erythroid Cells | https://www.ncbi.nlm.nih.gov/geo/query/acc.cgi?acc=GSE27602 | NCBI Gene Expression Omnibus, GSE27602 |
| Baena E, Shao Z | 2013 | Distinct transcriptional programs controlled by ERG and ETV1 in prostate cells | https://www.ncbi.nlm.nih.gov/geo/query/acc.cgi?acc=GSE39388 | NCBI Gene Expression Omnibus, GSE39388 |
| Huck-Hui Ng, Jia Chi Yeo, Jonathan Goke | 2014 | ChIP-seq of Klf2 in mouse embryonic stem cells (Serum/LIF and 2i) | https://www.ebi.ac.uk/arrayexpress/experiments/E-MTAB-2365/ | ArrayExpress Archive, E-MTAB-2365 |

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
