## [Decision Letter]

[**Editorial note:** This article has been through an editorial process in which the authors decide how to respond to the issues raised during peer review. The Reviewing Editor's assessment is that all the issues have been addressed.]

Thank you for submitting your article "Mitochondrial biogenesis is transcriptionally repressed in lysosomal lipid storage diseases" for consideration by *eLife*. Your article has been reviewed by three peer reviewers, and the evaluation has been overseen by a Reviewing Editor and Ivan Dikic as the Senior Editor. The following individual involved in review of your submission has agreed to reveal her identity: Julia Sellin (Reviewer #2). The other reviewers remain anonymous.

The Reviewing Editor has highlighted the concerns that require revision and/or responses, and we have included the separate reviews below for your consideration. If you have any questions, please do not hesitate to contact us.

The reviewers find the paper interesting but they all point out several short comings of this work. Some of the limitations, highlighted by the comments of reviewer 1 and especially reviewer 2, are of technical or statistical nature and need to be fully addressed. Furthermore, the rescue and requested protein levels experiments suggested in several places are important. Finally, critical experiments to substantiate the notion that alterations in the mitochondrial biogenesis impact disease should be performed.

More precise wording based on the reviewers’ suggestions is required to improve the manuscript. Thorough attention to the reviewers’ comments will greatly improve the manuscript.

Separate reviews (please respond to each point):

*Reviewer #1:*

In this paper, the authors show that the accumulation of transcription factors KLF2 and ETV1 represses mitochondrial biogenesis in Niemann-Pick disease patient fibroblasts. Using publicly available transcriptome data of Niemann-Pick type C model mouse tissues and in silico promoter analysis of respiratory chain genes, they discovered a link between KLF2, ETV1 and mitochondrial genes. Interestingly they discover that in sphingomyelinase deficient patient cells sphingosine-1-phosphate receptor 1 localization at plasma membrane is altered, resulting in a link between S1PR signaling and KLF2.

This article is interesting because it shows for the first-time that KLF2 and ETV1 are transcriptional repressors for mitochondrial biogenesis and that lysosomal dysfunction affects mitochondria via cellular signaling beyond autophagy. While certain conclusions are supported by the presented data, this manuscript requires further experimentation to confirm and clarify key results.

Specific points:

In this article, authors analyze two acid sphingomyelinase deficient patient cell lines. To clarify which cell line is analyzed in each experiment, authors are requested to name and specify them in figures. In Corcelle-Termeau et al., 2016, three NPA fibroblasts were analyzed. Authors should describe which cell line they analyzed in this paper. Is it possible to show mutation sites of the in-house NPA fibroblast?

In Figure 3, authors analyzed only a single cell line from an ASM-deficient patient. It is not clear whether the reduced OCR and mtDNA level are specific features of this cell line or general. Authors should analyze mitochondrial functions in other fibroblasts including GM18398 as in Figure 3. Rescue experiments with wild type sphingomyelinase and NPC1 are also needed to clarify causal relationships.

In Figure 3A, authors measured OCR by Seahorse and showed quantifications in the bar graphs. Although control and ASM cells supposedly responded to antimycin and rotenone, the observed values for the ASM cells are considerably lower than control cells. Authors need to discuss why non-mitochondrial respiration is lower in ASM cells. In order to identify mitochondrial OCR, non-mitochondrial respiration should be subtracted from other measurements (Figure 3B, Figure 3—figure supplement 3D, Figure 3—figure supplement 4D, 5D, 7D, and 7F).

In order to check if the respiratory chain defects are observed at the protein level, the authors should clarify the abundance of respiratory chain components by SDS-PAGE and Western blot analysis.

In the final paragraph of subsection “Impaired mitochondrial respiration in NPC and ASM deficiency”, authors claimed U18666A treatment results in accumulation of lysosomal sphingomyelin. However, a previous report (Baulies et al., 2015, Scientific Reports) showed U18666A did not affect lysosomal distribution of sphingomyelin. Authors are requested to show appropriate references for the comment. Authors also describe that desipramine induces mitochondrial dysfunction. Since desipramine is an inhibitor of both acid sphingomyelinase and acid ceramidase, the results seem not to support the conclusion that the effects of mitochondrial impairment is independent of acid ceramidase activity as described in the last sentences.

Authors found the enrichment of KLF2 and ETV1 in mitochondrial promoters by in silico analysis and analyzed ETV1 ChIp targets based on the data of Baena et al., 2013. In order to clarify the results, authors should perform ChIP-qPCR and luciferase reporter assay to verify the direct interaction of the transcription factors to some representative mitochondrial promoters.

In Figure 4Bb, authors observed the levels of KLF2 and ETV1 in NPC and ASM deficient fibroblasts by Western blotting and quantified the bands. As ETV1 was not altered in NPC fibroblasts even though KLF2 was up-regulated, authors should explain how the differences between NPC and ASM deficient fibroblasts arose. Silencing experiments of KLF2s are also required to clarify whether increased KLF2 represses mitochondrial biogenesis in NPC fibroblasts as Figure 5.

Authors described that KLF2 is upregulated post-translationally, but not transcriptionally, through deactivation of Akt in ASM deficient fibroblasts (Figure 6B). Akt transcriptionally regulates Klf2 expression in activated CD8^+^ T cells (Skon, Nat Immunol, 2013) and cited articles (Sinclair et al., 2008; Skon et al., 20013) did not show proteasomal degradation of KLF2 through Akt. Akt signaling can elicit proteasomal degradation of FoxO1, an inducer of Klf2 transcription (Plas, D.R. and Thompson, C.B, 2003, J. Biol. Chem.). Posttranslational regulation of KLF2 by Akt in Niemann-Pick fibroblasts needs to be analyzed properly.

Authors described KLF2 induces ETV1 through ERK signaling, however ERK activation by KLF2 shown in Figure 8A was not assessed in this article. In order to clarify whether KLF2 induces ERK activation, authors should test the ERK phosphorylation by KLF2 silencing in ASM cells.

In Figure 7G and H, authors show that an S1PR1 agonist leads to KLF2 inhibition, whereas S1PR1 inhibition increases KLF2 in control fibroblasts. Is ETV1 is also altered in control fibroblasts treated with Sew2871 and W146?

In Figure 8D, authors showed S1PR1 at the plasma membrane in ASM-deficient cells was less than control cells. Since they could detect S1PR1 and the reduction of S1PR1 by FTY720 treatment in ASM-deficient cells, they should revise the text of "in patient cells, S1PR1 is undetectable at the plasma membrane" in the Abstract.

Minor points:

1) Legend of Figure 3D is missing.

2) Kirkegaard et al., 2010 analyzed Niemann-Pick disease (NPD) A and B fibroblasts but not type C. Authors should cite appropriate references for NPC fibroblasts.

3) In Baena et al., Genes Dev, 2013, they took advantage of ChIP-on-chip with LNCaP cells, but not ChIpSeq. I would ask the authors to correct the figure legend of Supplementary Figure 9. Authors also should show the accession code of the dataset.

4) As two siRNA duplex sequences of KLF2 and ETV1, respectively, are shown in Supplementary file 4, authors should clarify which siRNAs they used in this research.

5) (B) and (F) in the legends of Figure 5 should be modified as (C) and (D).

6) References of Oninla et al., 2014, Lu et al., 2015, Zhu et al., 2016, and Cho et al., 2015 are missing. References of (53) and (54) in Materials and methods should be shown properly.

7) 50μM chloroquine (Σ, C6628) in Material and methods is not used in this article.

8) Nomenclature and mixed cases should be unified.

*Reviewer #2:*

In the present study, Yambire et al. discover that aberrant S1P signaling via S1PR1 and dysregulation of its downstream targets ETV1 and KLF2 cause mitochondrial abnormalities in Niemann-Pick animal models and patient cells (NPC1 and ASM mutants). Analysis of organelle specific gene lists in these mutants situations revealed, besides a to-be-expected lysosomal phenotype, an overall reduction of mitochondria-relevant genes, likely impacting on mitochondrial biogenesis and function. Further analysis yields the candidate transcription factors ETV1 and KLF2, which were subsequently analyzed in detail with respect to mitochondrial gene regulation and impact on mitochondrial function. A candidate approach to search for relevant regulators of ETV1 and KLF2 revealed the involvement of S1P signaling in mitochondrial phenotype induction and progression.

Overall, the study by Yambire et al. discovers a pathway contributing to NPC pathology and provides a novel explanation for mitochondrial abnormalities observed in this disease, which is an important field-specific contribution. Furthermore, their organelle gene list approach identifies the first known negative regulator of overall mitochondrial function, making the study interesting for a broader audience as well.

While I do not doubt the overall findings of the study and their relevance for the field, I have some concerns with regard to data presentation issues and unclear information about biological and technical replicates, especially in case of real time RT PCR results.

For example, in Figure 2A-D (real time data of mitochondrial gene expression in NPC and ASM mutant tissues (mice) and cells (patients)), it is stated that "one experiment out of two" is shown – what constitutes an experiment, and why is only one of those shown? Why not combine all biological replicates into one graph? Or is "one experiment" a technical replicate? What is N=8 in Figure 2A? Number of mice? What n is underlying Figure 2B? (three plates, therefore I assume n=3?). (should be n, not capitalized N, for n=sample size, which normally should represent biological replicates). What is the difference between "independent experiment" and "independent replicates" in the second sentence of the Figure 2 legend? Similar issues arise in other figures (e.g., Figure 4D). The Materials and methods section "Statistical Analysis" is also not quite clear on biological vs. technical replicates. Figure 3—figure supplement 1A does not give any information on sample size. Additionally, the use of standard deviation and s.e.m. changes between experiments, and it is unclear to the reader what the rationale between these choices is.

The presentation of real time PCR data was calculated as ΔΔCT (mentioned in the figure legend of Figure 2, not in the Materials and methods section, which I would have preferred). However, that method normally results in -fold change values (ΔΔCT = log2(fold-change)), which should be used in the y axis title (instead of "Gene expression/ reference gene"). -fold change values also mean that the control condition (in Figure 2A, that would be NPC1+/+) is automatically 1 (as the control condition is not changed relative to the control condition, therefore the -fold change is 1). A standard deviation of this value does therefore not make sense, unless all the data was normalized to one biological replicate (which is not explained in the legend, nor the Materials and methods section). In Figure 1B and C (and Figure 1—figure supplement 1), the y axis title states "Fold change (% of control)", which is a contradiction – it can be either -fold change, or% of controls. Furthermore, the graph seems to show neither of those two possibilities, but "difference to control in% ", which would explain the negative values. In Figure 4—figure supplement 3, a log FC (I assume log -fold change?) of overall gene expression is given in the figure – which would mean that the average -fold change for mitochondrial genes is 1.25fold and for respiratory chain 1.4fold, not 10% and 15% respectively, which is stated in the legend. The normalization to KLF2+/+ would indeed result in a log -fold change of 0 for KLF2+/+, which fits to the graph. Which one is correct, the graph or the legend?

Strictly speaking, HPRT and GAPDH are "reference genes", not "control genes" (e.g., figure legend of Figure 2).

In Figure 4C (quantification of bands), the normalized band intensity is given, but it is unclear to what it was normalized (I assume to the empty vector control band intensity?).

Figure 7 G and H are unclear to me – normalization to control would imply values relative to 1, not to 0 (band intensity (experiment)/band intensity (control)=0 would mean no band in experiment). What exactly is depicted? Band intensity (experiment) – band intensity (control), i.e., not-normalized values?

While the study is conducted in some depth, I would have liked to see analysis of Porin protein amounts or citrate synthase activity as an estimate of mitochondrial abundance/mass. Tfam expression and mtDNA only give a good estimate for mtDNA copy number, which not always correlates with mitochondrial mass and biogenesis. Secondly, it would be interesting to see where the S1PR1 receptor is located in ASM mutant cells with an antibody staining, as a relatively easy experiment to confirm and expand the FACS data (Figure 8D). Lastly, in subsection “KLF2 regulates ETV1 in an ERK-dependent manner”, AMPK activity is estimated by analyzing ACC phosphorylation as an AMPK target. I wonder why the authors did not check also for AMPK phosphorylation itself, as ACC phosphorylation is not always a direct measure for AMPK activity, as far as I know.

Sometimes there is a tendency to overstate results slightly. For example, in subsection “Silencing of KLF2 and ETV1 in ASM deficiency rescues mitochondrial biogenesis and function” the authors claim that protein levels of mitochondrial genes are rescued by ETV1 and KLF2 overexpression, but only Tfam was analyzed in this respect. In subsection “KLF2 regulates ETV1 in an ERK-dependent manner” (concerning Figure 6A), it is stated that ETV1 knockdown has no effect on KFL2 expression, which I do not agree with, although the effect is much smaller than the other way around. While this does not really change overall interpretation of the results, the change is statistically significant according to the graph.

I am a bit unclear about the paragraph in subsection “KLF2 and ETV1 are up-regulated in NPC1-/- tissues and repress transcription of mitochondria-associated genes”. What was the result of the IPA (in terms of what was determined – TFs that are differentially expressed in NPC1 mutant tissues, or relevant/enriched pathways with THEIR according TFs, i.e. TFs with predicted higher activity in the analyzed tissue?). In Figure 4A, the former is described ("Active/repressed TFs in NPC1-/- bain and liver", left circle), but the text implies the latter (subsection “KLF2 and ETV1 are up-regulated in NPC1-/- tissues and repress transcription of mitochondria-associated genes” paragraph two). The Materials and methods section is also not totally clear on this (statistically enriched in terms of what? Expression levels of the TFs? Number of pathway targets differentially expressed?).

Lastly, I was wondering why NRF1 (and 2) did not show up in the initial screening for candidates as described in Figure 4A. They should recognize cis-elements in mitochondrial genes, and I would have expected them to come up in an ingenuity pathway analysis. Maybe the authors could discuss this?

Minor Comments:

• Subsection “Expression of mitochondria-related genes is decreased in NPC1-/- tissues”: Hard to find the "gene lists", as they are not referenced the first time they are mentioned in the results

• Subsection “KLF2 and ETV1 are up-regulated in NPC1-/- tissues and repress transcription of mitochondria-associated genes”, final paragraph: "Given that ETV1 and KLF2 are predicted by our promoter analysis to have binding sites in.…" – to have binding sites implies the presence of these sites in the enhancer of ETV1 and KLF2 – I suggest "to recognize binding sites…".

• Subsection “KLF2 regulates ETV1 in an ERK-dependent manner”: Figure not cited (…as we have shown above, ETV1 is regulated at transcript level.…) (found in Figure 4—figure supplement 1)

• Subsection “KLF2 regulates ETV1 in an ERK-dependent manner”: ERK is not an effector (i.e. downstream), but a regulator of ETV1?

• Figure 1A: type very small, hard to read. The same is true for Figure 8A

• Figure 5 is numbered f instead of d and b instead of c

• Figure 7: It would be helpful to indicate in the figure itself that Sew2871 is an activator, and W146 an inhibitor of S1PR1 (in the title of a and b, for example). In subsection “S1PR1 signaling dynamically regulates KLF2 and mitochondrial biogenesis and function”, Figure 7 B is not cited in main text. I would prefer if all Sew2871 data would be in the left column and all W146 in the right column for easier grasp of concept.

• Table 1 was hard to find (in the pdf, it is not in the Materials and methods section, but at the end of the pdf).

• Accession number of KLF2-/- mouse dataset is missing in Materials and methods

• Figure 6A: asterisks for significance unclear (*** and ***** defined in legend, but there is ****** in figure – or is that 2x*** for two adjacent comparisons?)

• An explanatory description of Figure 9 would be helpful

Additional data files and statistical comments:

See major comments. I suggest to provide all source data for original experiments (mainly real time RT PCRs and WB band intensity plots, including uncropped Western blots as pictures).

*Reviewer #3:*

Yambire and colleagues address an underlying cause of the previously documented mitochondrial dysfunction that occurs in lysosomal storage diseases. They report the intriguing findings that in Niemann Pick Disease, deactivation of sphingosine receptor reduces mitochondrial biogenesis by activating the repressive transcription factors KLF2 and ETV1. The elucidation of a network of transcription factors that presumably tunes mitochondrial biogenesis (Nrf1 is the activator, KLF2/ETV1 are repressors) is certainly of interest, especially in the context of lysosomal storage disease. The authors demonstrate that during lysosomal stress, cells harbor fewer mitochondrial proteins with reduced activity. The data are clearly presented and the manuscript nicely written. I have a couple of modest concerns.

Does de-repression of KLF2/ETV1 affect any aspect of modeled Niemann Pick Disease? The authors write the manuscript as to suggest that increased mitochondrial biogenesis will improve aspects of disease. However, it may not. The KLF2 and ETV1 knockdown cells should be well-suited for such studies.

What is the relationship between S1P and mitochondria? Presumably the repression of mitochondrial biogenesis during S1PR impairment evolved for some reason other than to cause mitochondrial dysfunction in Niemann Pick Disease. Some additional discussion related to this would be appreciated.

[Editors' note: further revisions were suggested, as described below.]

Thank you for resubmitting your work entitled "Mitochondrial biogenesis is transcriptionally repressed in lysosomal lipid storage diseases" for further consideration at *eLife*. Your revised article has been favorably evaluated by Ivan Dikic (Senior Editor), a Reviewing Editor, and three reviewers.

The manuscript has been improved but there are some remaining issues and concerns with respect to statistics and data presentation that need to be addressed before publication, as outlined in the reviewer comments below:

• The statistics paragraph of Materials and methods is still very short and lacks some general considerations about the statistics used. It also does not cite the Cumming, Fidler and Vaux 2007 paper referenced in the point to point reply to my previous comments.

• In fact, Cumming, Fidler and Vaux 2007 do not encourage the usage of standard deviation. Furthermore, since almost all original experiments show rather small sample sizes (n=2-5) after clarification of biological vs. technical replicates, individual data points should be depicted, and the use of st.dev., s.em. and t-test is not appropriate, as is stated in Cumming, Fidler and Vaux (Rule 4) and many other guidelines on statistics for biologists. (It is not appropriate to use (inferential) statistics on technical replicates, as they do not influence sample size and all parameters calculated from n, like st. dev., s.e.m or p-value – which means that the sample sizes are in almost all experiments quite small, sometimes only n=2 or 3 in these cases, error bars and p-values are dubious.)

• Figure 1: as I have stated before, "fold change (% of controls)" does not make sense, as fold change means that the experiment is x-fold increased relative to control, i.e. a fold change of 2 means it is doubled, a fold change of 0.5 means its only half of the control condition. This cannot be expressed as% , which is of course also a term for change, but not for FOLD change. I suggest again to write "difference to control (%)". There is no explanation about error bar usage (Rule 1 in Cumming, Fidler and Vaux). There is no number of n given (Rule 2 in Cumming, Fidler and Vaux).

• Figure 1—figure supplement 1: see figure 1.

• Figure 4—figure supplement 3 (previous Figure 8) – figure legend is still incorrect – a log FC of 0.1 is NOT 10% increase! (10 to the power of 0.1 = 1.25-fold change, which means an increase of expression of +25%, or to 125% of control). I stated this in the first review and am disappointed that it is not corrected.

• Figure 4D (prev. 4C): I would prefer if the answer with respect to normalization from the point to point reply were also stated in the figure legend (band density, normalized to empty vector control).

Furthermore, we very strongly encourage you to provide original data, for example blot quantifications and numbers behind the charts to support the understanding and replication of the findings.

---

## [Author Response]

Reviewer #1:

[…] This article is interesting because it shows for the first-time that KLF2 and ETV1 are transcriptional repressors for mitochondrial biogenesis and that lysosomal dysfunction affects mitochondria via cellular signaling beyond autophagy. While certain conclusions are supported by the presented data, this manuscript requires further experimentation to confirm and clarify key results.

We have performed a number of new experiments and added several new pieces of data, detailed below.

Specific points:In this article, authors analyze two acid sphingomyelinase deficient patient cell lines. To clarify which cell line is analyzed in each experiment, authors are requested to name and specify them in figures. In Corcelle-Termeau et al., 2016, three NPA fibroblasts were analyzed. Authors should describe which cell line they analyzed in this paper. Is it possible to show mutation sites of the in-house NPA fibroblast?

We have now added the mutation sites of the in-house NPA fibroblasts. The figures now specify which results are from ASM-1 (the in-house NPA fibroblasts) or from ASM-2 (NPA fibroblasts obtained from Coriell Institute). We have also added in the Materials and methods all the cell lines used in this study.

In Figure 3, authors analyzed only a single cell line from an ASM-deficient patient. It is not clear whether the reduced OCR and mtDNA level are specific features of this cell line or general. Authors should analyze mitochondrial functions in other fibroblasts including GM18398 as in Figure 3. Rescue experiments with wild type sphingomyelinase and NPC1 are also needed to clarify causal relationships.

We have analyzed the key mitochondrial phenotypes in all ASM and NPC lines. Mitochondrial biogenesis and superoxide levels are shown in Figure 2, the OCR is shown in Figure 3. The results were similar to the ASM-1 line that we showed in the previous version. In addition, rescue experiments with ASMwt and NPC1wt, showed increased mitochondrial gene expression, associated with increased OCR in all the patient lines – these results are now included in Figure 3—figure supplement 1C-E.

In Figure 3A, authors measured OCR by Seahorse and showed quantifications in the bar graphs. Although control and ASM cells supposedly responded to antimycin and rotenone, the observed values for the ASM cells are considerably lower than control cells. Authors need to discuss why non-mitochondrial respiration is lower in ASM cells. In order to identify mitochondrial OCR, non-mitochondrial respiration should be subtracted from other measurements (Figure 3B, Figure 3—figure supplement 3D, Figure 3—figure supplement 4D, 5D, 7D, and 7F).

We thank the referee for noting this unclear point. The mitochondrial OCR calculations have been updated with subtraction of non-mitochondrial OCR. The real-time respirometry approach with the SeaHorse Extracellular Flux Analyzer typically results in a minor (usually negligible) O2 consumption rate when the respiratory chain is blocked with antimycin and rotenone. We have optimized the concentrations of the reagents for the SeaHorse assay (oligomycin, FCCP, antimycin and rotenone) for fibroblast cells, several years ago, and use always the same, for standardization purposes. So the protocol is not optimized for every single cell line, otherwise it wouldn’t be possible to compare them. As for the reason why there is a slight difference in non-mitochondria O2 consumption between ASM-1 and controls, we have not addressed that in particular. A possible explanation is that the concentrations of antimycin and rotenone used did not inhibit 100% of the activity of the respiratory chain, and since there is higher O2 consumption in control cells than ASM-deficient, there would still be a slight difference if only 5% or 1% of the electrons are passing through inhibited complexes I and III. It could also be that enzymes that consume O2 in the cytoplasm (e.g., dioxigenases such as prolyl hydroxylases, etc.) are less abundant or active in the ASM-deficient cells, or that enzymes that produce O2 (e.g., catalase) are more active.

In order to check if the respiratory chain defects are observed at the protein level, the authors should clarify the abundance of respiratory chain components by SDS-PAGE and Western blot analysis.

We thank the referee for raising this issue. We have performed several experiments to tackle this question, and have included new data on the protein levels of mitochondrial respiratory chain subunits in all lines tested. In addition, we verified mitochondrial mass by flow cytometry. These new results are shown in the Figure 3—figure supplement 2. Results from all lines irrespective of technique used, showed that mitochondrial mass is not reduced unlike at transcript level. The likely reason been that, as defective lysosomal storage disorders, the patient lines accumulate non-dysfunctional mitochondria due to the inability to complete mitophagy. We validated this reasoning by flow cytometry where we assessed the amount of mitotracker green (potential insensitive) vs red (functional mitochondria) and found significantly increased amount of dysfunctional mitochondria (mitotracker red negative but positive for mitotracker green) in all the patient lines. Importantly, in control fibroblasts without defective autophagy, manipulating S1PR1 activity and subsequent differential regulation of KLF2 and ETV1 levels, alters mitochondrial biogenesis both at transcript and at protein levels (Figure 7G-H). We have now addressed the discrepancy between mitochondrial biogenesis (transcript levels) and mitochondrial mass (result of the balance between decreased mitochondrial biogenesis and severely impaired mitophagy).

In the final paragraph of subsection “Impaired mitochondrial respiration in NPC and ASM deficiency”, authors claimed U18666A treatment results in accumulation of lysosomal sphingomyelin. However, a previous report (Baulies et al., 2015, Scientific Reports) showed U18666A did not affect lysosomal distribution of sphingomyelin. Authors are requested to show appropriate references for the comment. Authors also describe that desipramine induces mitochondrial dysfunction. Since desipramine is an inhibitor of both acid sphingomyelinase and acid ceramidase, the results seem not to support the conclusion that the effects of mitochondrial impairment is independent of acid ceramidase activity as described in the last sentences.

We thank the referee for noting that we were not clear on this point. We intended to state that there are biochemical overlaps between ASM and NPC disorders with secondary accumulation of cholesterol and sphingomyelin respectively as suggested previously in Platt, 2014, and Schuchman and Desnick, 2017. That notwithstanding, experimental evidence by Santos et al., 2015 showed reduced sphingomyelinase activity in rat astrocytes treated with U18666A. It is also worthy of note that Baulies et al., 2017 demonstrated convincingly that sphingomyelin homeostasis was not altered following 24h of U18666A treatment in primary mouse hepatocytes. Given that we treated human fibroblasts with U18666A for 72h, it is likely that the different models and times of treatment used, contributed to the discrepancies in our findings. Nevertheless, we have modified the sentence to make it less definite.

Authors found the enrichment of KLF2 and ETV1 in mitochondrial promoters by in silico analysis and analyzed ETV1 ChIp targets based on the data of Baena et al., 2013. In order to clarify the results, authors should perform ChIP-qPCR and luciferase reporter assay to verify the direct interaction of the transcription factors to some representative mitochondrial promoters.

We thank the referee for this suggestion. We have analyzed publicly-available Klf2 ChIP-seq data (arrayexpress E-MTAB-2365; Yeo et al., 2014). We found a large number of KLF2 targets (i.e., that meet the significance cut-off for the ChIP-seq analysis) in genes that are included in the “mitochondrial gene list”. These are now shown in Figure 4—figure supplement 4.

In Figure 4B, authors observed the levels of KLF2 and ETV1 in NPC and ASM deficient fibroblasts by Western blotting and quantified the bands. As ETV1 was not altered in NPC fibroblasts even though KLF2 was up-regulated, authors should explain how the differences between NPC and ASM deficient fibroblasts arose. Silencing experiments of KLF2s are also required to clarify whether increased KLF2 represses mitochondrial biogenesis in NPC fibroblasts as Figure 5.

We thank the referee for bringing this point up. We work with other transcription factors (TF) in the lab, and often the total protein level of the TF is not necessarily indicative of activity because of shuttling between the nucleus and the cytoplasm. For that reason, we prepared nuclear and non-nuclear “cytoplasmic” extracts, to test if KLF2 and ETV1 were enriched in the nucleus. We find that there is a clear increase of KLF2 and ETV1 in the nucleus of ASM-1 cells (now shown in Figure 4C), but also in the nucleus of ASM-2 (now shown in Figure 4—figure supplement 1) and NPC cells (now shown in Figure 4—figure supplement 1D).

We also silenced KLF2 and ETV1 in NPC fibroblasts (now shown in Figure 5F), and observed the expected increase in the expression of mitochondrial genes (now shown in Figure 5G).

Authors described that KLF2 is upregulated post-translationally, but not transcriptionally, through deactivation of Akt in ASM deficient fibroblasts (Figure 6B). Akt transcriptionally regulates Klf2 expression in activated CD8^+^ T cells (Skon, Nat Immunol, 2013) and cited articles (Sinclair et al., 2008; Skon et al., 20013) did not show proteasomal degradation of KLF2 through Akt. Akt signaling can elicit proteasomal degradation of FoxO1, an inducer of Klf2 transcription (Plas, D.R. and. Thompson, C.B, 2003, J. Biol. Chem.). Posttranslational regulation of KLF2 by Akt in Niemann-Pick fibroblasts needs to be analyzed properly.

We thank the referee for pointing out this unclear point. We have revised the text accordingly to avoid suggesting that Akt directly targets KLF2 for proteosomal degradation. Pursuing the exact mechanism of KLF2 post-translational regulation is a very interesting question, which we are working on, but this manuscript already covers so many aspects that we feel adding another layer of detail in this particular subject would decrease the readability of the manuscript.

Authors described KLF2 induces ETV1 through ERK signaling, however ERK activation by KLF2 shown in Figure 8A was not assessed in this article. In order to clarify whether KLF2 induces ERK activation, authors should test the ERK phosphorylation by KLF2 silencing in ASM cells.

We thank the referee for this comment, which results from lack of clarity in the way we described the KLF2-ETV1 interaction. We don’t intend to suggest that KLF2 induces ERK activation, at least we do not yet have the experimental evidence to say so. Rather, we imply that the induction of ETV1 by KLF2 requires ERK activation, which when inhibited (U0126) in Figure 6B abrogates ETV1 induction by KLF2. We have modified the text accordingly.

In Figure 7G and H, authors show that an S1PR1 agonist leads to KLF2 inhibition, whereas S1PR1 inhibition increases KLF2 in control fibroblasts. Is ETV1 is also altered in control fibroblasts treated with Sew2871 and W146?

This is an excellent point. We have now included data on ETV1 and VDAC1 levels in Sew2871 and W146 treated fibroblasts. ETV1 behaves in complete agreement with KLF2, and VDAC behaves in agreement with the mitochondrial proteins (opposite to KLF2 and ETV1).

In Figure 8D, authors showed S1PR1 at the plasma membrane in ASM-deficient cells was less than control cells. Since they could detect S1PR1 and the reduction of S1PR1 by FTY720 treatment in ASM-deficient cells, they should revise the text of "in patient cells, S1PR1 is undetectable at the plasma membrane" in the Abstract.

Again, thank you for pointing out our incorrect wording. The text has been revised accordingly, we now state that S1PR1 is barely detectable in patient cells.

Minor points:1) Legend of Figure 3D is missing.

Corrected accordingly.

2) Kirkegaard et al., 2010 analyzed Niemann-Pick disease (NPD) A and B fibroblasts but not type C. Authors should cite appropriate references for NPC fibroblasts.

Point noted and corrected.

3) In Baena et al., Genes Dev, 2013, they took advantage of ChIP-on-chip with LNCaP cells, but not ChIpSeq. I would ask the authors to correct the figure legend of Figure 4—figure supplement 4. Authors also should show the accession code of the dataset.

Reviewer’s point is noted and corrected accordingly. Accession code for datasets has been included.

4) As two siRNA duplex sequences of KLF2 and ETV1, respectively, are shown in Supplementary file 4, authors should clarify which siRNAs they used in this research.

Both siRNA duplexes were combined for efficient knockdowns.

5) (B) and (F) in the legends of Figure 5 should be modified as (C) and (D).

Legends have been updated.

6) References of Oninla et al., 2014, Lu et al., 2015, Zhu et al., 2016, and Cho et al., 2015 are missing. References of (53) and (54) in Materials and methods should be shown properly.

References now updated

7) 50μM chloroquine (Σ, C6628) in Material and methods is not used in this article.

Point well noted, and chloroquine is taken out of the article.

8) Nomenclature and mixed cases should be unified.

We have now unified the nomenclature.

Reviewer #2:

[…] While I do not doubt the overall findings of the study and their relevance for the field, I have some concerns with regard to data presentation issues and unclear information about biological and technical replicates, especially in case of real time RT PCR results.

We thank the referee for pointing out that the legends were not clear. We have now revised the legends accordingly, as detailed below.

For example, in Figure 2A-D (real time data of mitochondrial gene expression in NPC and ASM mutant tissues (mice) and cells (patients)), it is stated that "one experiment out of two" is shown – what constitutes an experiment, and why is only one of those shown? Why not combine all biological replicates into one graph? Or is "one experiment" a technical replicate? What is N=8 in Figure 2A? Number of mice? What n is underlying Figure 2B? (three plates, therefore I assume n=3?). (should be n, not capitalized N, for n=sample size, which normally should represent biological replicates). What is the difference between "independent experiment" and "independent replicates" in the second sentence of the Figure 2 legend? Similar issues arise in other figures (e.g., Figure 4D). The Materials and methods section "Statistical Analysis" is also not quite clear on biological vs. technical replicates. Figure 3—figure supplement 1A does not give any information on sample size. Additionally, the use of standard deviation and s.e.m. changes between experiments, and it is unclear to the reader what the rationale between these choices is.

We thank the referee for this comment. We agree that the legends were over-complicated and unclear. Legends have been revised to clarify this confusing description. All the N= are now shown as n= and indeed they correspond to a biological replicate (mouse, or cell plate grown independently). Independent replicates means different biological samples (e.g., eight mice are eight different biological replicates; three biological replicates for cells means three plates of cells, collected and processed independently, e.g., three protein extracts or three RNA extracts, etc..). Regarding the standard deviation and the standard error of the mean, we follow the guidelines by Cumming, Fidler and Vaux (2007), J. Cell Biol.. Specifically, for experiments in which we have a moderate number of biological replicates and technical replicates, we use the standard deviation, because it is a descriptive error bar. However, when the sample/replicate number is higher (we set an arbitrary number of at least 20), such as in the analysis of the gene lists (n is in the hundreds) or when using plate-based assays (plate reader, respirometry), we have a large number of technical replicates, and in these cases the mean of the result is much closer to the mean of the population, and thus we use inferential error bars, i.e., the standard error of the mean.

The presentation of real time PCR data was calculated as ΔΔCT (mentioned in the figure legend of Figure 2, not in the Materials and methods section, which I would have preferred). However, that method normally results in -fold change values (ΔΔCT = log2(fold-change)), which should be used in the y axis title (instead of "Gene expression/ reference gene"). -fold change values also mean that the control condition (in Figure 2A, that would be NPC1+/+) is automatically 1 (as the control condition is not changed relative to the control condition, therefore the -fold change is 1). A standard deviation of this value does therefore not make sense, unless all the data was normalized to one biological replicate (which is not explained in the legend, nor the Materials and methods section).

qCPR data analyses now moved to Materials and methods section and text revised for clarity. Indeed, when performing a qPCR we have multiple biological replicates of the control samples, we set one of them as reference sample and normalize all the other samples (including the other control replicates) to that sample – that is where the standard deviation comes from. This is now clarified in the Materials and methods.

In Figure 1B and C (and Figure 1—figure supplement 1), the y axis title states "Fold change (% of control)", which is a contradiction – it can be either -fold change, or% of controls. Furthermore, the graph seems to show neither of those two possibilities, but "difference to control in% ", which would explain the negative values. In Figure 4—figure supplement 3, a log FC (I assume log -fold change?) of overall gene expression is given in the figure – which would mean that the average -fold change for mitochondrial genes is 1.25fold and for respiratory chain 1.4fold, not 10% and 15% respectively, which is stated in the legend. The normalization to KLF2+/+ would indeed result in a log -fold change of 0 for KLF2+/+, which fits to the graph. Which one is correct, the graph or the legend?

Indeed, logFC means log of the fold change. The text has been rewritten for clarity.

Strictly speaking, HPRT and GAPDH are "reference genes", not "control genes" (e.g., figure legend of Figure 2).

HPRT and GAPDH are now referred to as reference genes in text.

In Figure 4C (quantification of bands), the normalized band intensity is given, but it is unclear to what it was normalized (I assume to the empty vector control band intensity?).

Normalized to empty vector control.

Figure 7 G and H are unclear to me – normalization to control would imply values relative to 1, not to 0 (band intensity (experiment)/band intensity (control)=0 would mean no band in experiment). What exactly is depicted? Band intensity (experiment) – band intensity (control), i.e., not-normalized values?

Point noted and revised as normalized band density (difference from vehicle treatment).

While the study is conducted in some depth, I would have liked to see analysis of Porin protein amounts or citrate synthase activity as an estimate of mitochondrial abundance/mass. Tfam expression and mtDNA only give a good estimate for mtDNA copy number, which not always correlates with mitochondrial mass and biogenesis. Secondly, it would be interesting to see where the S1PR1 receptor is located in ASM mutant cells with an antibody staining, as a relatively easy experiment to confirm and expand the FACS data (Figure 8D). Lastly, in subsection “KLF2 regulates ETV1 in an ERK-dependent manner”, AMPK activity is estimated by analyzing ACC phosphorylation as an AMPK target. I wonder why the authors did not check also for AMPK phosphorylation itself, as ACC phosphorylation is not always a direct measure for AMPK activity, as far as I know.

This point is well taken, and responded to above since reviewer 1 also had similar concerns. AMPK phosphorylation has been included in the data in Figure 6B.

Sometimes there is a tendency to overstate results slightly. For example, in subsection “Silencing of KLF2 and ETV1 in ASM deficiency rescues mitochondrial biogenesis and function” the authors claim that protein levels of mitochondrial genes are rescued by ETV1 and KLF2 overexpression, but only Tfam was analyzed in this respect.

We stated that TFAM was just an example of such rescue following ETV1 and KLF2 silencing (not overexpression). In the figures, we also show that NRF1 levels are rescued. Given the confounding effect of defective mitochondrial clearance (Figure 3—figure supplement 2) and the fact that ETV1 and KLF2 silencing does not alter defective autophagy in these models (Figure 5—figure supplement 1), assessing mitochondrial proteins directly was found to be not useful. However, in Figure 7, in the absence of the confounding effect caused by impaired autophagy, it is clearly shown that all respiratory chain subunits tested behave similarly to observed changes in transcript levels.

In subsection “KLF2 regulates ETV1 in an ERK-dependent manner” (concerning Figure 6A), it is stated that ETV1 knockdown has no effect on KFL2 expression, which I do not agree with, although the effect is much smaller than the other way around. While this does not really change overall interpretation of the results, the change is statistically significant according to the graph.

We revised the wording, and it is now stated as negligible effect.

I am a bit unclear about the paragraph in subsection “KLF2 and ETV1 are up-regulated in NPC1-/- tissues and repress transcription of mitochondria-associated genes”. What was the result of the IPA (in terms of what was determined – TFs that are differentially expressed in NPC1 mutant tissues, or relevant/enriched pathways with THEIR according TFs, i.e. TFs with predicted higher activity in the analyzed tissue?). In Figure 4A, the former is described ("Active/repressed TFs in NPC1-/- bain and liver", left circle), but the text implies the latter (subsection “KLF2 and ETV1 are up-regulated in NPC1-/- tissues and repress transcription of mitochondria-associated genes” paragraph two). The Materials and methods section is also not totally clear on this (statistically enriched in terms of what? Expression levels of the TFs? Number of pathway targets differentially expressed?).

The text has now been revised for clarity. The TFs identified by IPA are filtered based on statistically over-represented targets in our gene lists, using Fisher’s exact test. Depending on the known effect of the TF on each target (activation/repression), it is then calculated the Z-score, which determines if the TF is predicted to be active or repressed. The TFs themselves are not necessarily differentially expressed, since many of these proteins are not regulated at transcript level, so we didn’t use this as a filter criterion.

Lastly, I was wondering why NRF1 (and 2) did not show up in the initial screening for candidates as described in Figure 4A. They should recognize cis-elements in mitochondrial genes, and I would have expected them to come up in an ingenuity pathway analysis. Maybe the authors could discuss this?

Cis-elements for NRF1 scored in many respiratory chain genes, but not in many others, and didn’t meet the statistical cut-off (Fisher’s exact test). While it is usually assumed that NRF1 triggers the expression of all mitochondrial genes, that is a notion that was extrapolated out of a few genes encoding mitochondrial proteins, and that does not necessarily hold true for the whole mitochondrial proteome, or even for the whole respiratory chain. NRF2 (GABPA) had less cis-elements than NRF1 in the respiratory chain subunits promoters. We didn’t discuss this in the text because we took un unbiased approach, and therefore to discuss specifically NRF1 or NRF2 would leave out several other TFs that trigger some aspects of mitochondrial biogenesis. We are addressing the transcriptional regulation of mitochondrial biogenesis in a systematic manner in the context of another project.

Minor Comments:• Subsection “Expression of mitochondria-related genes is decreased in NPC1-/- tissues”: Hard to find the "gene lists", as they are not referenced the first time they are mentioned in the results.

Gene list referenced sooner for clarity.

• Subsection “KLF2 and ETV1 are up-regulated in NPC1-/- tissues and repress transcription of mitochondria-associated genes”, final paragraph: "Given that ETV1 and KLF2 are predicted by our promoter analysis to have binding sites in.…" – to have binding sites implies the presence of these sites in the enhancer of ETV1 and KLF2 – I suggest "to recognize binding sites…".

Point noted and revised as suggested.

• Subsection “KLF2 regulates ETV1 in an ERK-dependent manner”: Figure not cited (…as we have shown above, ETV1 is regulated at transcript level.…) (found in Figure 4—figure supplement 2).

This point is well noted and Figures have been cited again and again for clarity.

• Subsection “KLF2 regulates ETV1 in an ERK-dependent manner”: ERK is not an effector (i.e. downstream), but a regulator of ETV1?

Point noted. ERK is indeed a regulator.

• Figure 1A: type very small, hard to read. The same is true for Figure 8A

Revised.

• Figure 5 is numbered F instead of D and B instead of C.

Updated.

• Figure 7: It would be helpful to indicate in the figure itself that Sew2871 is an activator, and W146 an inhibitor of S1PR1 (in the title of a and b, for example). In subsection “S1PR1 signaling dynamically regulates KLF2 and mitochondrial biogenesis and function”, Figure 7 B is not cited in main text. I would prefer if all Sew2871 data would be in the left column and all W146 in the right column for easier grasp of concept.

Figures rearranged, and 7B cited.

• Table 1 was hard to find (in the pdf, it is not in the Materials and methods section, but at the end of the pdf).

Tables have been rearranged accordingly.

• Accession number of KLF2-/- mouse dataset is missing in Materials and methods

All accession numbers for datasets used in this study have now been included.

• Figure 6A: asterisks for significance unclear (*** and ***** defined in legend, but there is ****** in figure – or is that 2x*** for two adjacent comparisons?)

Well noted! 2x*** for adjacent comparisons, revised accordingly.

• An explanatory description of Figure 9 would be helpful.

Graphic representation of publicly available Etv1 ChIP-chip data, highlighting how many of its target genes encode proteins involved in different aspects of mitochondrial function. Etv1 target genes from the ChIP-seq data were crossed with the mitochondrial ‘gene list’ described above to obtain targets of Etv1 which encode mitochondrial proteins. We note that NRF1 and PPARG, although they do not encode bona fide mitochondrial proteins, they were found to be direct targets of Etv1 and thus their inclusion.

Reviewer #3:

Yambire and colleagues address an underlying cause of the previously documented mitochondrial dysfunction that occurs in lysosomal storage diseases. They report the intriguing findings that in Niemann Pick Disease, deactivation of sphingosine receptor reduces mitochondrial biogenesis by activating the repressive transcription factors KLF2 and ETV1. The elucidation of a network of transcription factors that presumably tunes mitochondrial biogenesis (Nrf1 is the activator, KLF2/ETV1 are repressors) is certainly of interest, especially in the context of lysosomal storage disease. The authors demonstrate that during lysosomal stress, cells harbor fewer mitochondrial proteins with reduced activity. The data are clearly presented and the manuscript nicely written. I have a couple of modest concerns.

We thank the referee for the positive comments on our manuscript.

Does de-repression of KLF2/ETV1 affect any aspect of modeled Niemann Pick Disease? The authors write the manuscript as to suggest that increased mitochondrial biogenesis will improve aspects of disease. However, it may not. The KLF2 and ETV1 knockdown cells should be well-suited for such studies.

We thank the referee for bringing up this important point. We have addressed it with new experiments. Although the text may give that impression, it was not our intention to suggest that increasing mitochondrial biogenesis will alleviate aspects of NP disease. On the contrary, we aimed at clarifying the involvement of mitochondrial and lysosomal crosstalk in the pathophysiology of the disease. Interestingly, we indeed find that increasing mitochondrial biogenesis via KLF2 and ETV1 silencing in ASM and NPC1 models is not beneficial since it increases apoptosis with a subsequent reduction in cell viability (shown in Figure 8E-G). These findings suggest that the induction of KLF2 and ETV1 in ASM and NPC1 is a protective adaptive mechanism to promote survival with a compromising repression of mitochondrial biogenesis.

What is the relationship between S1P and mitochondria? Presumably the repression of mitochondrial biogenesis during S1PR impairment evolved for some reason other than to cause mitochondrial dysfunction in Niemann Pick Disease. Some additional discussion related to this would be appreciated.

The role of S1P signaling in mitochondrial biogenesis does not pertain only to NP disease. Indeed, in hepatocytes, S1P signaling was found to be necessary for mitochondrial biogenesis via PKA-PGC1a. Furthermore, S1P signaling in T cells was shown to be essential for the maintenance of mitochondrial content and function. As well as our results, these findings suggest that S1P signaling might be a previously underappreciated regulatory loop for mitochondrial biogenesis. Given that S1P signaling was shown to engage PGC1a, and we have shown for the first time that lack of its signaling engages repressors of mitochondrial biogenesis, S1P signaling might be a crucial denominator for maintaining mitochondrial content. We have now addressed this question in the discussion.

[Editors' note: further revisions were suggested, as described below.]

The manuscript has been improved but there are some remaining issues and concerns with respect to statistics and data presentation that need to be addressed before publication, as outlined in the reviewer comments below:• The statistics paragraph of Materials and methods is still very short and lacks some general considerations about the statistics used. It also does not cite the Cumming, Fidler and Vaux 2007 paper referenced in the point to point reply to my previous comments.

We have included further details in the statistical description in the Materials and methods. We must note that we provide information on the statistical treatment for every panel, in the respective figure legend. We have also included clarification on biological replicates and their technical replicates thereof.

• In fact, Cumming, Fidler and Vaux 2007 do not encourage the usage of standard deviation. Furthermore, since almost all original experiments show rather small sample sizes (n=2-5) after clarification of biological vs. technical replicates, individual data points should be depicted, and the use of st.dev., s.em. and t-test is not appropriate, as is stated in Cumming, Fidler and Vaux (Rule 4) and many other guidelines on statistics for biologists. (It is not appropriate to use (inferential) statistics on technical replicates, as they do not influence sample size and all parameters calculated from n, like st. dev., s.e.m or p-value – which means that the sample sizes are in almost all experiments quite small, sometimes only n=2 or 3 – in these cases, error bars and p-values are dubious.)

We respectfully disagree with reviewer #2 on this instance. Cumming, Fidler and Vaux neither encourage nor discourage the use of standard deviation, although they suggest that standard errors or confidence intervals can be used instead. It is also standard practice in biology to present the standard deviation, including papers signed by referee #2 (Buelow et al., MBoC 2017 and Sellin et al., PLOS Biology 2018). We must respectfully note that in this manuscript error bars are mainly presented as s.e.m. in the graphs (as opposed to the consistent miscommunication in the initial figure legends in which we identified as SD. We have therefore accurately updated the figure legends to address this pertinent concern. In support of this, we are including an image (Author response image 1) for Figure 2A,C to illustrate the point that the error bars represent s.e.m. and not SD.

However, we must point out that we disagree with the conclusion of the referee regarding the number of biological replicates. We are using patient cells. So in the most extreme interpretation, everything is a technical replicate, because it comes from n=1 patient. That notwithstanding, we define biological replicates as follows: for in vivoexperiments n refers to the number of mice for each genotype used in this study and for all cell culture experiments we state that n refers to the number of independent experiments carried out with different stocks of each cell line. We also included several technical replicates for each biological experiment. This definition is standard and prescribed by others including Cumming, Fidler and Vaux. In this case, the sample size for almost all cell culture experiments in this manuscript is at least n=3, which is commonplace for such work. Despite the perceived reduced number of samples, we employed many different cell lines from the same and related genetic defects, to sustain the validity of our biological conclusions. We show that by using pharmacological approaches the same conclusions are reached. We show that by using mouse models (n=8 or 9), the same conclusions are reached. How will the same dubious conclusions be arrived at using several models and approaches? Besides, the instances with n=2 involve the robust and highly reproducible (as evidenced by the technical replicates) real-time respirometry experiments with the Seahorse FluxAnalyzer where pharmacological inhibition of ASM and NPC were carried out. We don’t find it necessary to include more replicates in all the experiments just for the sake of statistical satisfaction without recourse to the biology.

For this particular point, we choose to take advantage of the possibility associated with the “Research Communications” in which the authors decide if and how to reply to the referees’ comments. We did not apply this to the rest of the comments in the first review and in the current one, but in this particular instance we choose not to further engage in what we consider a discussion detached from the biological conclusions of our study.

• Figure 1: as I have stated before, "fold change (% of controls)" does not make sense, as fold change means that the experiment is x-fold increased relative to control, i.e. a fold change of 2 means it is doubled, a fold change of 0.5 means its only half of the control condition. This cannot be expressed as% , which is of course also a term for change, but not for FOLD change. I suggest again to write "difference to control (%)". There is no explanation about error bar usage (Rule 1 in Cumming, Fidler and Vaux). There is no number of n given (Rule 2 in Cumming, Fidler and Vaux).

We corrected the text to “Fold change (difference from WT)”. The number of genes analyzed is alluded to in the legend, and clearly detailed in the organelle proteome table and the number of biological samples (mice) is now included in the figure legends.

• Figure 1—figure supplement 1: see Figure 1.

Since the same datasets were used for the analyses, the number of biological samples is the same as stated above and corrected in the figure legends.

• Figure 4—figure supplement 3 – figure legend is still incorrect – a log FC of 0.1 is NOT 10% increase! (10 to the power of 0.1 = 1.25-fold change, which means an increase of expression of +25%, or to 125% of control). I stated this in the first review and am disappointed that it is not corrected.

We respectfully note this comment and we have corrected it accordingly.

• Figure 4D: I would prefer if the answer with respect to normalization from the point to point reply were also stated in the figure legend (band density, normalized to empty vector control).

This point is noted and included in the Materials and methods section for experiments in which we normalized the results of experimental samples to those of the corresponding controls to ensure easy comparisons.